# Physical vitrification and nanowarming at liter-scale CPA volumes: toward organ cryopreservation

Lakshya Gangwar[1], Zonghu Han [1], Cameron Scheithauer [1],
Bat-Erdene Namsrai[2], Saurin Kantesaria[3], Rob Goldstein[4], Michael L. Etheridge [1],
Erik B. Finger [2,6,7] & John C. Bischof [1,3,5,6,7]

Organ banking via vitrification could transform transplantation, but has never been achieved at human organ scales. This study tested vitrification and rewarming in 0.5–3 L volumes using cryoprotective agents (CPAs): M22, VS55, and 40%EG + 0.6 M Sucrose. Ice formation and cracking was avoided through optimized convective cooling, and successful vitrification was confirmed via visual inspection, thermometry, and X-ray μCT. M22 and EG+sucrose vitrified at 0.5 L, but only M22 succeeded at 3 L; VS55 failed at all volumes. Porcine livers (~0.6–1 L total volume; ~0.23–0.75 L organ volume) were also vitrified using EG+sucrose, though not rewarmed. Future experiments are needed to optimize the protocol and achieve liver rewarming. Using nanowarming with iron-oxide nanoparticles and a newly developed 120 kW RF coil, uniform rewarming was achieved in up to 2 L volumes of M22 at ~88 °C/min. This work serves as a proof-of-concept that human organ scale vitrification and rewarming is physically possible, thereby enabling human organ banking in the future.

Vitrification, or the rapid cooling of biologic material to an ice-free glassy state at ultralow temperature, promises indefinite storage of cells, tissues, and organs for transplantation or other biomedical applications[1,2]. Over a century ago, the first successful vitrification of frog spermatozoa was conducted by Luyet and Hodapp (in 1938)[3,4]. Subsequently, by the 1980s, vitrification was performed on multiple biological systems, including whole rabbit kidneys (10's of mL volumes)[1]. Since then, vitrification has been attempted up to 1.5 L but failed due to underlying ice formation and fracture[5], inherent to heat transfer at these scales. Even under conditions where vitrification upon cooling was successful, even faster rates are required upon rewarming, leading to failures (ice formation and/or cracking) at the rabbit kidney scale[6]. The critical cooling rate (CCR) and critical warming rate (CWR) are the required cooling and warming rates needed for vitrification and rewarming without ice formation.

Several volumetric rewarming techniques have since been tested to overcome the limitations of conventional boundary layer rewarming, including high-intensity focused ultrasound (HIFU) and dielectric/microwave warming. HIFU heats by focusing sound/pressure waves on the vitrified material but is currently limited to volumes ≤2 mL[7]. It also has the inherent limitation of wave penetration, reflections, acoustic impedance mismatch at interfaces, cavitation, and the possibility of thermal overshoot in the liquid phase[8,9]. Dielectric/microwave rewarming (heating of dipolar molecules by electromagnetic (EM) waves) potential rewarming volumes can approach ~100 mL in CPA systems[10], including rabbit kidneys (~50 mL)[11]. However, this approach also has several limitations, including small penetration depths, poor EM coupling in the vitrified "solid" state, and the potential for "thermal runaway" near the melt[12], each of which are accentuated by the

[1]Department of Mechanical Engineering, University of Minnesota, Minneapolis, MN, USA. [2]Department of Surgery, University of Minnesota, Minneapolis, MN, USA. [3]Department of Biomedical Engineering, University of Minnesota, Minneapolis, MN, USA. [4]AMF Life Systems, LLC, Auburn Hills, MI, USA. [5]Institute for Engineering in Medicine, University of Minnesota, Minneapolis, MN, USA. [6]These authors contributed equally: Erik B. Finger, John C. Bischof. [7]These authors jointly supervised this work: Erik B. Finger, John C. Bischof. ✉e-mail: efinger@umn.edu; bischof@umn.edu

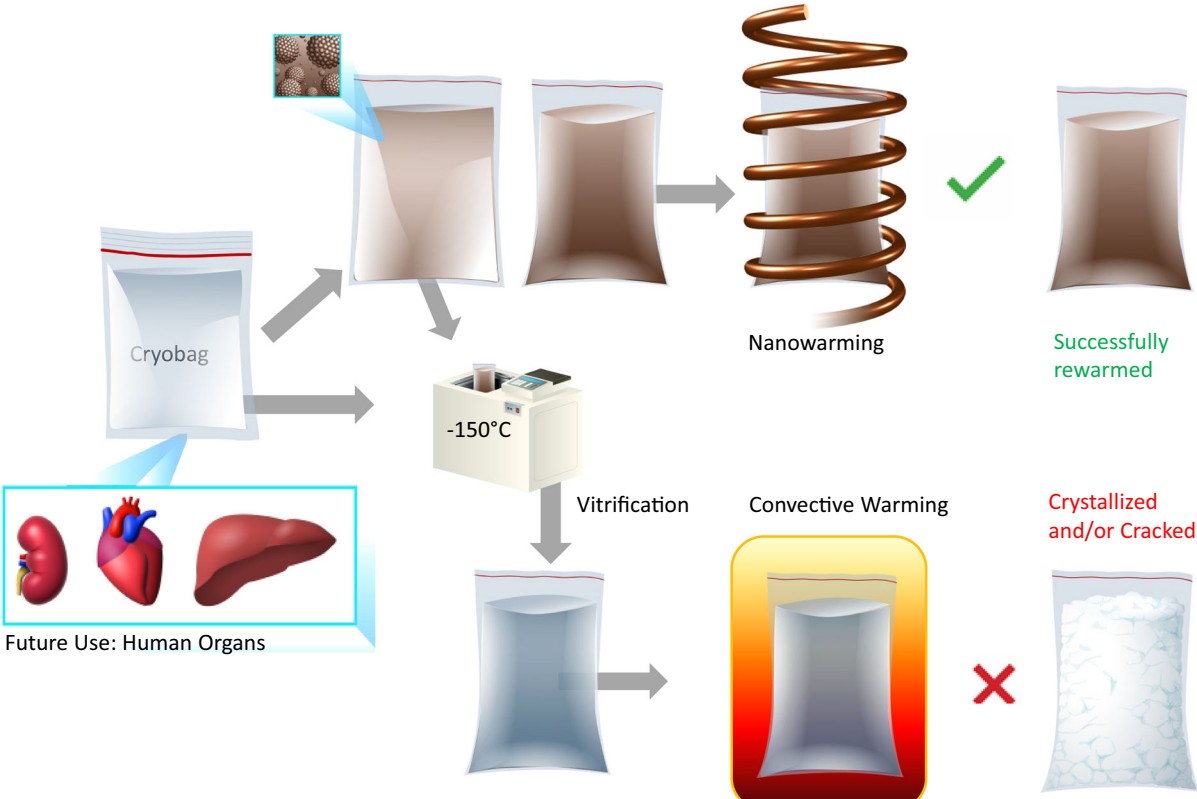

**Fig. 1 | Schematic flow of steps (left to right) in liter-scale vitrification and rewarming.** Liter volumes of a CPA (0.5–3 L) in cryobags are large enough to hold a human organ. The cryobag is placed inside a controlled rate freezer (CRF) for cooling. For nanowarming (top section of the flow chart), the cryobag is vitrified with iron oxide nanoparticles (IONPs) suspended in the CPA. The vitrified cryobag is placed inside the RF coil, alternating magnetic fields are turned on, and the IONPs generate heat. This leads to successful (rapid, uniform) rewarming, avoiding crystallization or cracking failure modes even at the liter scale. Traditional rewarming employs convection using a water bath, which at these scales results in ice recrystallization and/or fractures due to insufficient rewarming rates (slow) and thermal stresses (non-uniform), respectively (lower section of the flow chart). IONP iron oxide nanoparticles, CPA cryoprotective agent, CRF controlled rate freezer.

inhomogeneity typical of biological tissues (and their associated dielectric properties), all of which are expected to worsen as the scale increases.

To address these limitations, our group developed a volumetric rewarming technology termed "nanowarming"[6,13], where magnetic iron-oxide nanoparticles (IONPs) are perfused through the organ vasculature before vitrification. For rewarming, the organs are placed in an alternating magnetic field that couples to the nanoparticles and induces heating from within the vasculature through magnetic hysteresis. Since the chosen radiofrequency waves penetrate the system without significant attenuation, this approach is, in theory, fully scalable to human organs. So far, we and others have published on nanowarming of volumes up to 80 mL CPA mL and 30 mL biological nanowarming at rates up to 50–100 °C/min[14–19] (see Table S1).

To achieve cryopreservation of human organs through vitrification, we anticipate needing 0.5–1 Liter volumes for hearts and kidneys, while human livers may require ≥3 L[20]. However, neither successful vitrification nor rewarming has been demonstrated in the literature at these scales. Since capillary spacing is relatively conserved across organs and tissues, uniformity and heating rate depend primarily on the concentration of IONPs, distribution of the magnetic field strength, and frequency. Given the timescales of rewarming, heating is practically uniform if the magnetic field is uniform, as the heat rapidly diffuses across intercapillary distances.

There are currently no volumetric means of cooling human organs, so convective cooling will be directly impacted by the size of the system, with the cooling rate at the center of the sample decreasing as the size of the system increases[21]. This may lead to insufficient cooling rates and/or physical fractures/cracking due to thermal gradients in the vitrified state[22] if a proper cooling protocol is not employed. It is also important to note that since nanowarming rates are independent of sample volume, cooling conditions will become the rate-limiting step in vitrifying and rewarming human organ-scale volumes.

Here, we describe the proof-of-concept for successfully achieving physical vitrification and nanowarming in human organ scale volumes. Figure 1 shows the sequential steps for this multi-liter vitrification and CPA nanowarming in a cryobag. We report the successful vitrification of volumes up to 3 L and uniform nanowarming from the vitrified cryogenic state of up to 2 L. We further demonstrate physical vitrification in a CPA-perfused liver (~1 L porcine liver). This study demonstrates that bulk systems equivalent in size to human organs can be vitrified and nanowarmed, as verified by visual inspection, thermometry, and μCT. Hence, this study is a physical demonstration towards human-scale organ cryopreservation.

## Results

### Modeling liter scale vitrification

Computational heat transfer modeling was performed to develop the vitrification protocols of 0.5, 1, and 3 L cryobag systems using COMSOL 5.4. Convection within the CPA itself is neglected due to the high viscosity of CPAs at subzero temperatures. Therefore, the general form of the governing equation for cooling and warming is assumed to be the conduction heat equation:

$$\rho C_P \frac{\partial T}{\partial t} = \nabla \cdot (k \nabla T) + q_v'''$$ (1)

Where ρ is density, $C_p$ is specific heat, k is thermal conductivity, T is temperature, and $q'''_v$ is volumetric heating (generated from IONP = volumetric specific absorption rate (SARᵥ)). Limited data on CPA thermal properties are available, hence we assume them to be that of M22 (listed in Table S2)[23], which should be a reasonable approximation for each of the other CPAs. The initial CRF temperature was set to 0 °C (see Fig. S4). Boundary cooling at h = 100 W/m²K (h: heat transfer coefficient) and $T_{CRF}$-−40 °C/min (maximum CRF cooling rate) was initiated, and upon reaching a temperature of −122 °C, just above $T_g$, the system was annealed (thermally equilibrated to minimize thermal gradients/stress, Fig. S4). Two important modeling outcomes studied were the cooling rate (dT/dt) in the region of ice formation (above ~ −100 °C), and the temperature difference ($\Delta T = T_{center}-T_{edge}$) in the glassy region (below $T_g$: glass transition temperature) where the system is at most risk of fracture/cracking failure. The characteristic length for heat transfer is defined as $L_C$ (=Volume/Surface Area)[24], which can be used to compare the trends in thermal predictions across different volumes (see Fig. S4). Note that larger volumes (and $L_C$) result in slower cooling rates and require longer annealing time (Table 1).

### Table 1 | Summary of cooling protocols developed based on heat transfer modeling for each vitrification volume

| CRF Cooling Protocol | 0.5 L | 1 L | 3 L |
|---|---|---|---|
| Characteristics Thermal Length: $L_C$ | 1.2 cm | 1.4 cm | 2.2 cm |
| Start Temperature | 0°C | 0°C | 0°C |
| Chamber initial cooling rate (to −122°C) | −40°C/min | −40°C/min | −40°C/min |
| Anneal (hold) at −122°C | 180 min | 250 min | 520 min |
| Chamber cooling rate in glassy region (from anneal to −150 °C) | −0.5 °C/min | −0.4 °C/min | −0.2 °C/min |
| Hold at −150°C | 35 min | 50 min | 90 min |
| Total Protocol Time (hrs) | 4.6 | 6.2 | 12.5 |
| Modeled Center Cooling Rate (0 to −100 °C) | 1.36 | 0.96 | 0.45 |

After annealing, the system was cooled slowly (<1 °C/min) to −150 °C for storage to minimize thermal stress, keeping the temperature difference (<20 °C) in the brittle glassy phase[21,25]. The summary of vitrification success for all 3 volumes and CPAs is shown in Fig. 2 (and Fig. S10). The temperature distribution across the geometry is plotted for the three CPA volumes (Figs. 3, S3 and S12). The minimum predicted and measured cooling rate (CR) (which is at the center) decreased with increased volumes from -1.4 °C/min for 0.5 L ($L_C$-1.2 cm) to -1 °C/min for a 1 L ($L_C$-1.4 cm) and finally -0.5 °C/min for 3 L ($L_C$-2.2 cm) cryobag.

## Measuring liter scale vitrification

Based on these cooling protocols (Fig. 3C), we then cooled cryobags containing three different CPAs (M22, VS55, 40%EG + 0.6Msucrose) at the three modeled volumes (0.5, 1, and 3 L). The success or failure of vitrification in these systems was verified by visual, thermal, and μCT measurements. Visually, ice could be identified as round spherulites during crystallization (Fig. S8A) or by opaque, milky-white/cloudy appearance throughout the sample in case of complete crystallization (center of Fig. S8B, S8C). Another mode of physical failure is fractures or cracking, which could be visually observed as the presence of linear defects (Fig. S9). In the absence of failure modes (crystallization and fractures), the CPAs looked clear, transparent, and glassy, implying successful vitrification (Fig. 2). In the case of organs, failure was assessed visually on the surface and internally by photos of the bisected vitrified organ to evaluate for ice. Vitrified material is more radiodense (higher Hounsfield Units, HU) on μCT than ice, and cracks could be detected directly by abrupt changes in radiodensity[14,17] (Fig. S13).

Successful vitrification in M22 was achieved for all volumes tested (Fig. 2). 40%EG + 0.6Msucrose was also successfully vitrified with no visual ice formation up to 1 L (Fig. S10). Ice formation was observed in VS55 for all three volumes (Fig. S8 and S10). This was expected as the achieved cooling rate for these three volumes (0.5 L-1.4 °C/min, 1 L-1 °C/min, 3 L-0.5 °C/min) were lower than the CCR of VS55 (-2.5 °C/min). We further confirmed our visual findings using μCT, which showed successful vitrification in M22 and 40%EG + 0.6Msucrose but

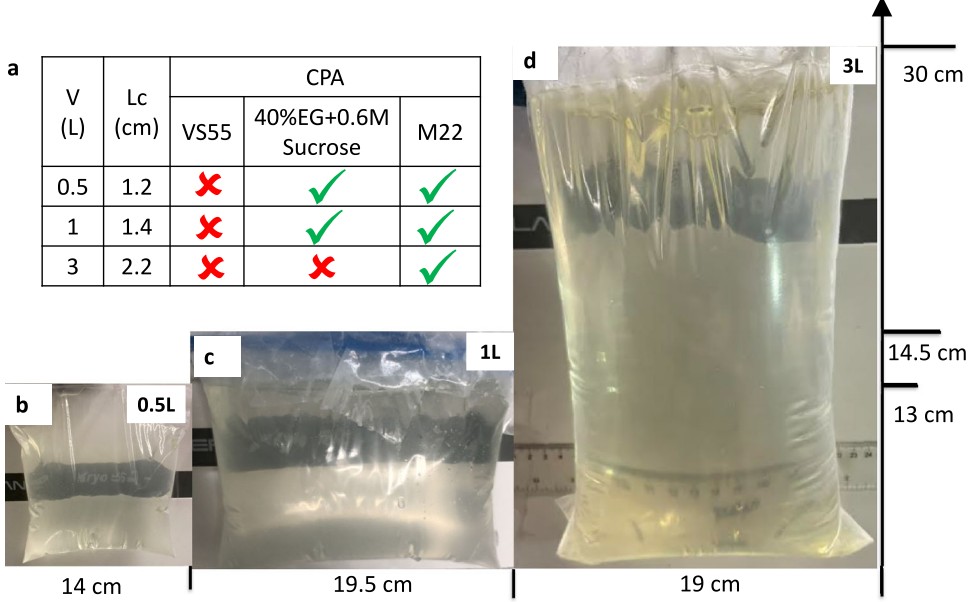

**Fig. 2 | Demonstration of physical success of vitrification in multiple volumes. a** Table summarizing vitrification results for all the three CPAs and volumes. Photos of successfully vitrified (glass) M22 inside a cryobag for **b** 0.5 Liter, **c** 1 Liter, and **d** 3 Liter (largest volume reported). The out-of-plane thicknesses are 5.5, 6.5, and 10.5 cm for 0.5, 1, and 3 L cryobags, respectively.

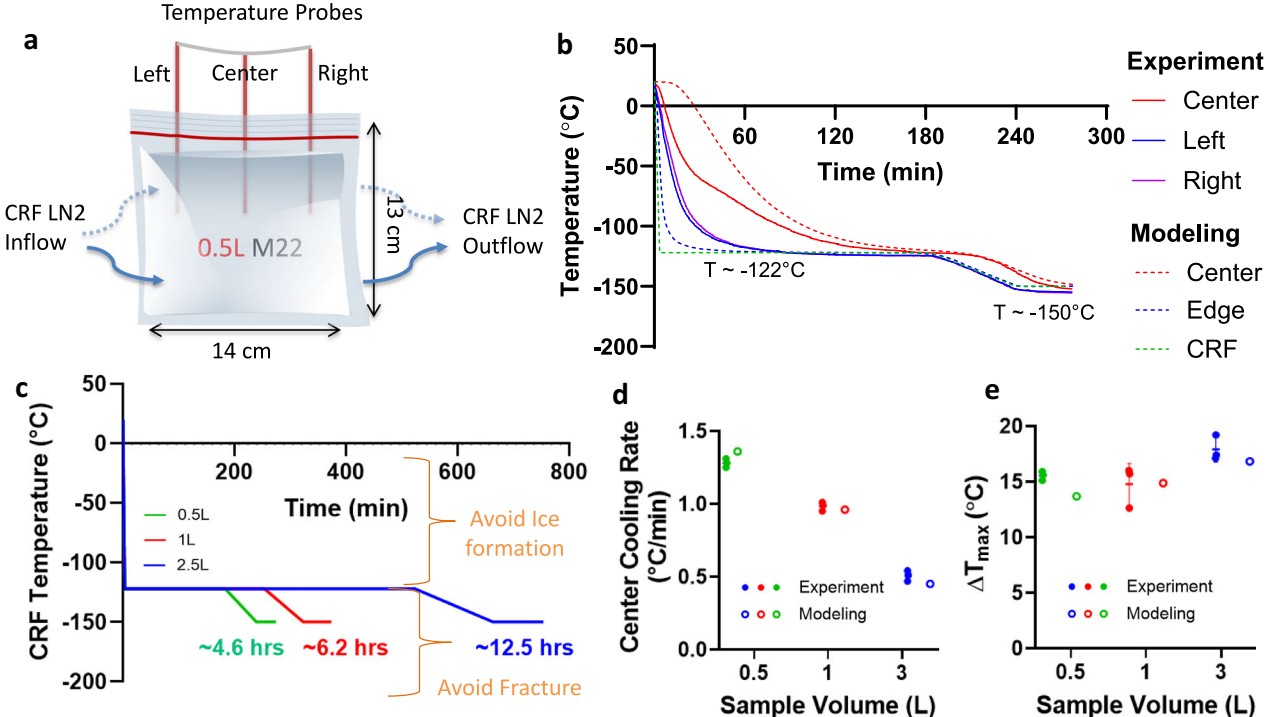

**Fig. 3 | Thermal results from experimental and modeled liter scale CPA vitrification. a** Schematic for a representative case, 0.5 L cryobag containing CPA with placement of three fiber optic temperature probes (3 cm apart). Blue arrows show the direction of LN2 flow in CRF. **b** Experimental and predicted temperature vs. time plot for 0.5 L M22. The dashed green line shows the programmed CRF temperature profile/protocol. **c** CRF cooling protocols for 0.5, 1, and 3 L volumes. The regions of ice formation and fracture danger are labeled. Scatter plot of **d** center cooling rate and **e** temperature difference ($\Delta T_{max}$ in the glassy region) for all three volumes tested for M22 (mean ± SD; $n = 3$ independent experiments from distinct samples). Cooling rate is calculated in ranges 0 to −100 °C and −120 to −150 °C for temperature difference plots. Mean cooling rates are greater than the CCR of M22 (-0.1 °C/min). Temperature differences are within the allowable limit (<20 °C) calculated from a simple thermal shock equation[21].

ice formation in VS55 at 0.5 L volumes (Fig. S13). The temperature profiles, cooling rates, and temperature differences are shown in Fig. 3 (and Fig. S12). As before, the cooling rates decreased, and the temperature differences increased with increasing sample volume.

### Vitrification of a human-scale organ - porcine liver

To test the feasibility of vitrifying an organ at these scales, we perfused porcine livers of varying size (inset table in Fig. S14) (similar in size to a juvenile, age ~ 10-year-old, liver) with 40%EG + 0.6Msucrose. Notably, these livers have greater volume than most other adult human organs (Table S6). The livers were vitrified in a cryobag with a surrounding CPA volume of ~ 200–400 mL. The 0.5 L CRF cooling protocol (Fig. 3C) was used for the liver because the estimated $L_C$ (-0.9 cm) of the porcine liver in CRF was closer to the $L_C$ (-1 cm) of the representative 0.5 L cryobag volume (see Fig. S7). This highlights the importance of appropriately designing cooling protocols for a given geometry. For instance, different shapes and form factors can be attained for a given volume resulting in different characteristic lengths, $L_C$. This difference in $L_C$ will change the convective cooling process for the system in the CRF, such as the cooling rate above $T_g$, annealing time, slow cooling rate in glassy region, and storage hold time (see Fig. S4 and S7 and supplementary section on porcine heat transfer modeling). The resulting vitrified porcine liver is shown in Fig. 4. The liver appeared vitrified based on visual inspection, except for a small amount of ice formed around the portal vein and fatty tissue (poorly vascularized perihilar tissue), which presumably did not equilibrate fully with the CPA. The vitrified liver was also bisected, and cross-sectional photos (Fig. 4) largely appeared to be free of ice. μCT images further confirms the absence of ice inside the liver tissue (Fig. S15). Modeling predicted the center cooling rate of the liver as -4 °C/min, which exceeds the CCR

(-1 °C/min) of 40%EG + 0.6Msucrose, which further supports vitrification (see Fig. S6).

Effective cryobag sealing was beneficial in preventing open-surface ice formation and reducing the size of the system ($L_C$). Vacuum sealing with the removal of surrounding CPA solution led to ice on superficial surfaces of the liver, potentially due to cryobag surface nucleation, so having a layer of CPA around the liver in the bag proved beneficial for successful vitrification.

### Characterization of liter scale RF rewarming coil

We next evaluated the ability to uniformly rewarm liter-scale volumes from the cryogenic vitrified state (<−120 °C). Although past studies have rewarmed whole rat and rabbit organs using a 15 kW RF coil (AMF Life Systems)[14,17,19] or other commercial RF coils[16,18], these studies were typically limited to <30 mL, with only a few nearing ~100 mL volumes (Table S1)[6]. As noted earlier, uniformity in rewarming will depend on the uniformity of the applied field, assuming practically uniform IONP distributions (i.e., through vascular perfusion to the capillary beds). Therefore, to achieve uniform nanowarming in clinical-scale volumes, we worked with AMF LifeSystems, LLC. to design, build, and characterize a 120 kW RF coil with a 2.5 L uniform field region. This substantially extends our capabilities beyond previous nanowarming studies (Fig. 5).

Detailed characterization of the alternating magnetic field within the RF coil was performed across the volume of interest in the new 120 kW system (Figs. S16–18) and compared to previous coil systems[6,26]. We measured the magnetic field using a 2D RF probe coil (AMF Lifesystems, LLC.) as a function of spatial distribution and applied coil voltage. At full power, the RF coil generated up to 35 kA/m magnetic field strength across the coil's volume at a frequency of

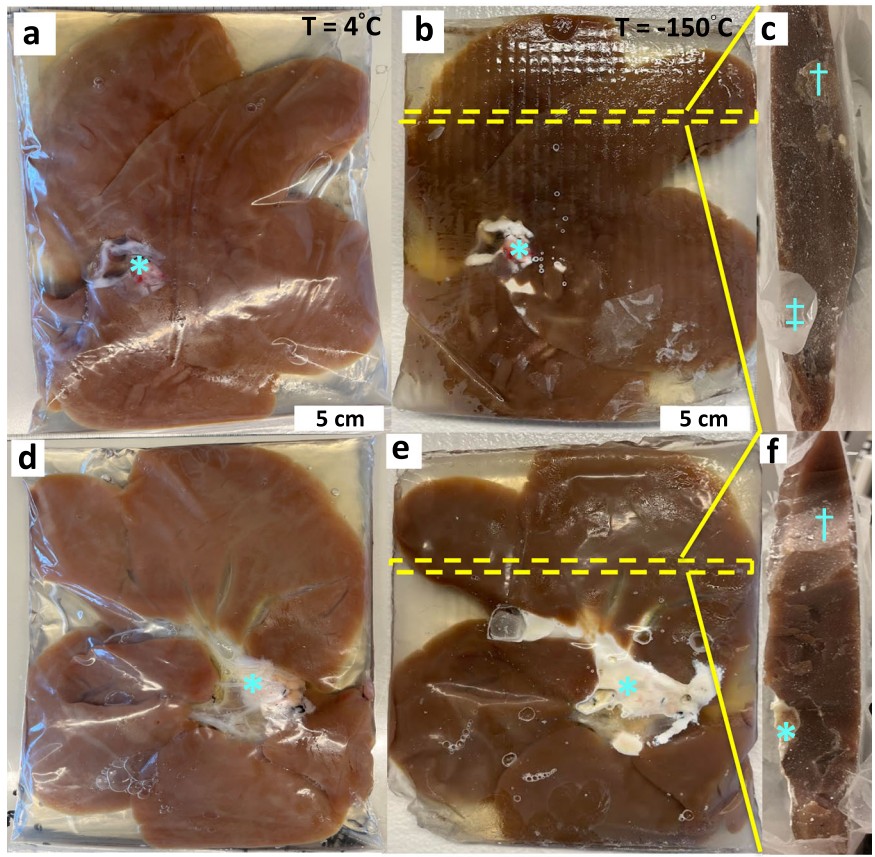

**Fig. 4 | Porcine Liver Vitrification.** Photos of a representative porcine liver **a**, **d** (left) before (T = 4°C) and **b**, **c**, **e**, **f** (right) after vitrification (T = −150 °C). The pattern in the vitrified liver was due to the cryobag placement on a supporting mesh in the control rate freezer (CRF) (see Fig. S7B). The cryobag was removed for the vitrified liver photo in **b**, **e** to reduce glare and get a clear photo. Rightmost photos **c, f** of a center cross-section of vitrified bisected pig liver. Yellow dashed line in **b**, **e** represents the location of an orthogonal bisection in liver. Blue (*) indicates perihilar and fatty tissue whereas blue (‡) denotes the cryobag. Blue (†) denotes some surface ice condensation that occurred between storage and photography in the bottom bisected liver.

360 kHz. The calibration showed a linear relationship between applied power and field strength (Fig. 5C) for the 120 kW coil. The magnetic field was spatially uniform (within 5%) for a 20 cm axial distance and 6 cm radial direction (Fig. 5D). Importantly, the maximum field strength and frequency output were comparable between the 15 kW and 120 kW systems. Additional power in the larger RF coil was required to sustain this field uniformly across the substantially higher coil volume (~80 mL versus 2.5 L).

To best design nanowarming protocols for use in the new system, we systematically evaluated IONP heating across the range of available field strengths, frequencies, and temperatures. To accomplish this, we measured the specific absorption rate ($SAR_{Fe}$, normalized to IONP mass) for IONPs in CPA (M22) as a function of magnetic field and frequency at both room and cryogenic temperatures. As seen in Fig. 6A, for the IONPs tested, SAR increased with magnetic field strength, though it saturated at higher field strengths, as observed elsewhere[27]. $SAR_{fe}$ increased at higher frequencies (~2 times more at 360 kHz than 190 kHz for 35 kA/m), suggesting a proportional increase in heating with frequency, consistent with prior reports[26,28,29]. Given that the output field is capped by the power available to the system (P ∝ freq. x H), proportionally higher rewarming is expected at higher frequency and field strength (e.g., 360 kHz and 35 kA/m vs. 180 kHz and 60 kA/m). Lastly, we report that $SAR_V$ increased at cryogenic temperature (by ~ 1.5 times) compared to room temperature (Fig. 6B). Further details of the SAR calculation can be found in the supplementary information (also see Fig. S19 for SAR of uncoated IONPs: EMG308).

## Liter scale nanowarming

To demonstrate nanowarming scales relevant for vitrified human organs, we prepared M22 with IONPs at ~10.7 mgFe/mL for 1 L and ~4.6 mgFe/mL for 2 L volume. Different cryobags than those used in the vitrification studies were used to fit in with the workable volume of our 120 kW RF coil (Fig. 7A, also see Figs. S21, S24, Table S3). After the 1 L and 2 L volumes were vitrified in the CRF following the same protocols described above, they were stored in a −150 °C freezer overnight. For rewarming, the samples were rapidly transferred to (within ~3–5 seconds) and rewarmed inside the 120 kW RF coil. Samples were rewarmed to 0°C within ~1 min for 1 L and ~2 min for 2 L at rates of ~172 °C/min and ~88 °C/min, respectively (Fig. 7C). This difference is due to the use of ~50% lower IONP concentration in the larger 2 L bag. Rates were identical when 0.5 L and 1 L volumes were rewarming at the same IONP concentration (Fig. S22). In all cases, the measured temperature differences across the cryobag volume were negligible (~<5°C) (Fig. 7D). Note that the vitrified porcine liver was not nanowarmed due to RF coil dimension restriction (diameter of coil being smaller than horizontal flat liver dimensions; also see Fig. S24).

As a final proof of principle, we nanowarmed a higher concentration of IONP (100 mgFe/mL EMG308) using VMP as the CPA (closely related to M22) in a 1 mL cryovial[30]. We achieved warming rates at ~1500 °C/min (Fig. S22). This is the fastest nanowarming rate that has been measured that we are aware of, supporting prior observations that rewarming rates will scale linearly with IONP concentration[29,31,32], which is promising for future development.

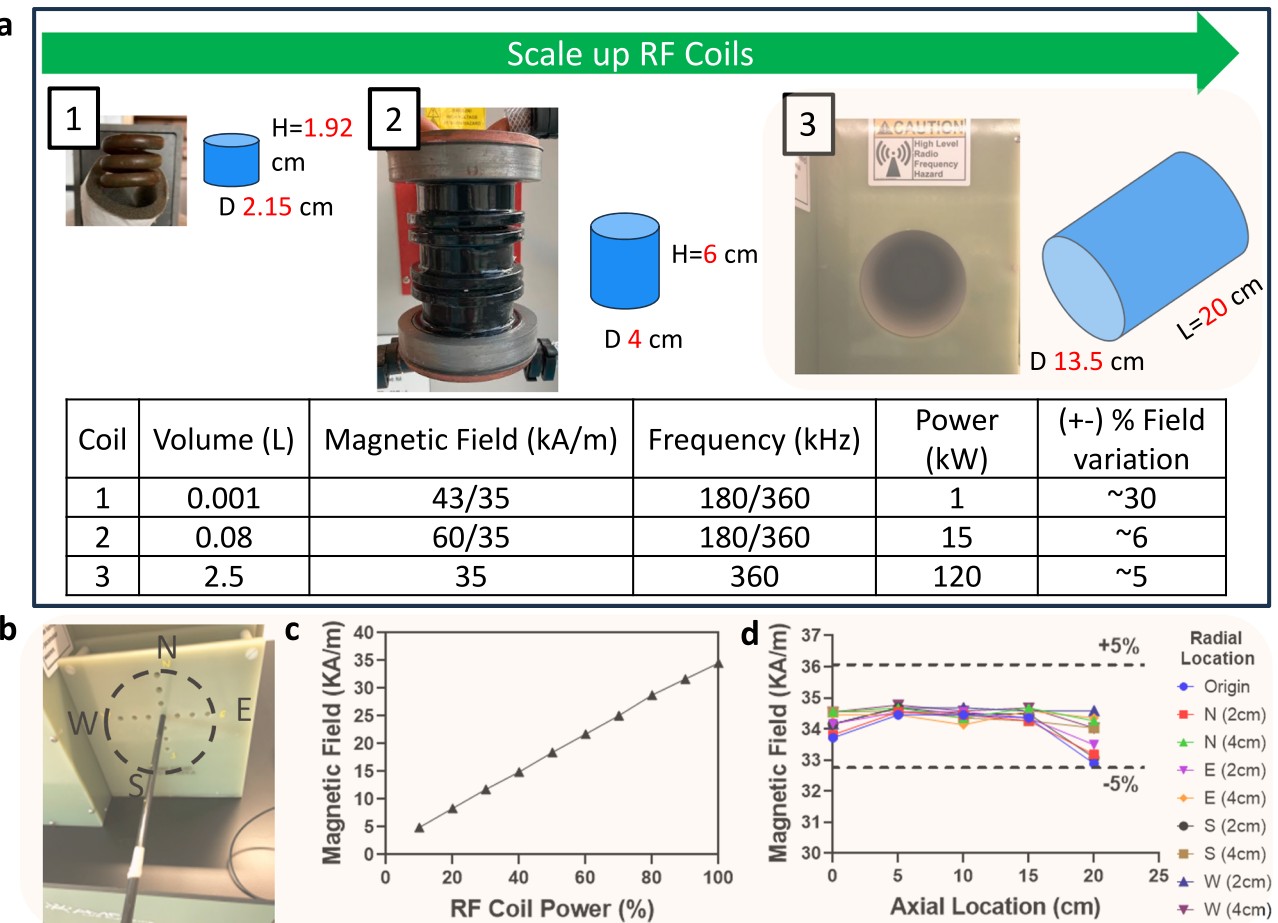

**Fig. 5 | Comparison of several generations of radiofrequency (RF) coil systems and characterization of 120 kW RF coil for liter-scale nanowarming. a** Photo of the RF coil system is shown alongside the schematic of the coil's approximate uniform volume (blue cylindrical region) with diameter (D) and length (L) listed next to blue region (not to scale). The bottom table compares operating power, magnetic field strength, frequency, sample volume, and uniformity of the systems. Note that the RF coil uniformity is a % variation of magnetic field strength across the RF coil volume (blue cylinders). The top green arrow shows the increasing sample scale for RF coils with practical capacities, such as 1 mL cryovials in a 1 kW coil, human ovaries and eyes in a 15 kW coil, and human kidney and heart in a 120 kW coil. **b** Photo of the RF coil bore and RF probe placement for magnetic field measurements with labels showing relative radial directions (North (N), South (S), East (E), and West (W)). **c** Plot of measured magnetic field strength vs. coil power. **d** Plot of magnetic field strength vs. axial location for different radial locations. The 120 kW system frequency is fixed at 360+/− 5 kHz.

## Discussion

As a useful first-order approximation, the minimum cooling rates to avoid ice formation during the vitrification of different human organs can be predicted based on their characteristic length ($L_C$) (Fig. 8A, Table S6, supplementary section on cooling rate calculation). This can be used to establish CPA concentration and cooling protocol requirements (Fig. 8B, see supplementary for calculating min. vitrifiable CPA concentration). For instance, a small organ such as an ovary can convectively cool at ~15 °C/min, whereas the center of a liver is only expected to cool at ~0.6 °C/min (these calculations assume the organ is vitrified alone without any surrounding CPA, but $L_C$ could be adjusted for individual container geometries). Note that all predicted cooling rates are greater than those achieved experimentally in the successfully vitrified 3 L volume of M22 (cooling at ~0.45 °C/min). For simplicity, these calculations are performed for organs without any surrounding volume (see supplementary section on estimation of surrounding CPA volume and Fig. S23).

For this study, we chose CPAs with low CCRs (<1 °C/min)[30,33]. However, during CPA perfusion loading of organs, successfully perfused tissues may only equilibrate to ~92-94% concentration of full-strength CPA[15,34], which will substantially increase the required cooling rates. Thus, a conservative approach would be to select a CPA where the CCR at ~94% loading is lower than the achievable cooling rate for the volume. For instance, using Figs. 8B, for 3 L we can calculate $L_C$ ~ 2.2 cm and the expected CR in the CRF to be <0.8 °C/min. Hence, the minimum CPA concentration for vitrification would be ~62% w/w, which is slightly lower than M22 (~66%w/w which includes carrier solution), where we have shown successful vitrification at 3 L. Higher concentrations of CPAs such as VS83 (83% w/w CPA) have even lower CCR and can be more easily vitrified but increase biological toxicity relative to the CPAs chosen here[35]. To remain at a lower concentration of CPA and still achieve vitrification at higher volumes without toxicity, future work can assess the impact of ice recrystallization inhibitors (IRIs), polymers (e.g., polyglycerol-PGL, polyvinyl alcohol-PVA, polyethylene glycol-PEG, x-1000, z-1000, etc.), or other novel cryoprotective agents[36,37]. Furthermore, due to larger heat transfer coefficients, convective cooling with liquid cryogens versus gaseous flow in CRF can enhance cooling rates[24].

The other major mode of physical injury during organ vitrification is cracking. To address this, it is crucial to thermally equilibrate samples through annealing, thereby reducing thermal gradients and stress before entering the glassy phase. Without proper annealing, these samples will be more susceptible to fracture during cryogenic storage and rewarming. Once annealed above $T_g$ (where the supercooled CPA

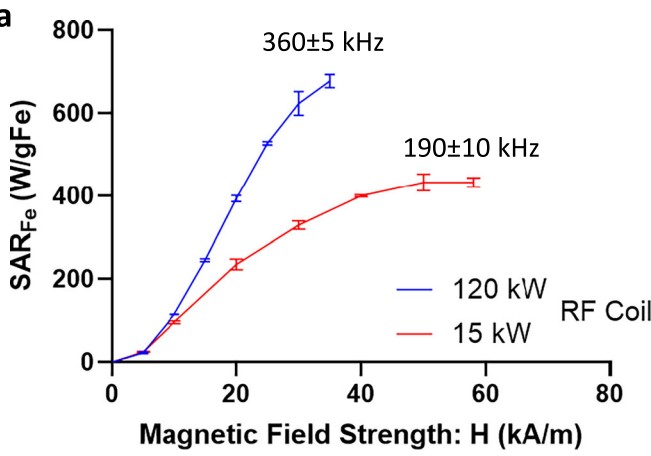

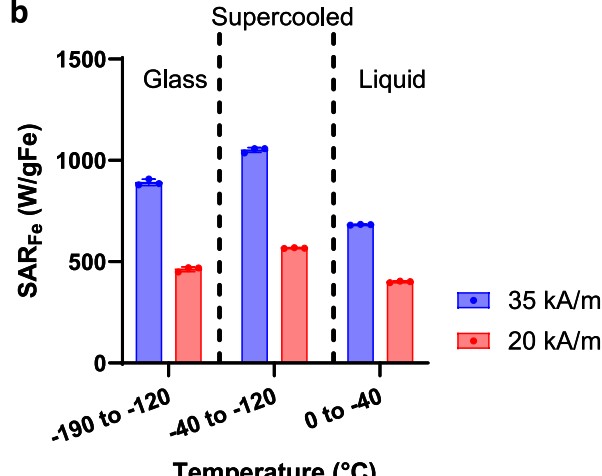

**Fig. 6 | Nanowarming specific absorption rate (SAR). a** Plot of $SAR_{Fe}$ ($SAR_V/C_{Fe}$) vs. magnetic field strength (H) measured at room temperature for iron-oxide nanoparticles IONPs (sIONPs in M22 shown here) at two frequencies (190 and 360 kHz) (plotted mean ± SD; n = 3 independent experiments from distinct samples). **b** Plot of $SAR_{Fe}$ vs. temperature for sIONPs in M22. Average $SAR_{Fe}$ (mean ± SD; n = 3, independent experiments from distinct samples) is plotted in three different temperature regions, i.e., glass, supercooled, and liquid. SAR is measured from cryogenic temperature (−196 °C) to room temperature (20 °C) at two different field strengths (20, 35 kA/m) and 360 kHz (see Fig. S19 for more details).

is fluid enough to relieve stresses), it is also critical to cool slowly enough to the storage temperature to avoid re-introducing significant thermal gradients and stress when cooling through the glass transition. To leverage these results for broader use and develop effective CRF cooling protocols (choosing annealing time, CR in the glassy region, equilibration time at storage, etc.) for volumes other than the ones analyzed in this study, we provide recommended annealing and glassy region CR as a function of $L_C$ as plotted in Fig. S4. These protocols were designed to be faster than the CCR of the noted CPAs between the melt and glass transitions. Further, the $\Delta T$ within the glassy state was designed to be less than 20 °C, to avoid fracture. For larger volumes, the extended cooling protocols (i.e., -12 hrs for 3 L) highlight the need for innovation, perhaps through a combination of convective and volumetric cooling approaches previously attempted in the kidney[38] and intestine[39].

While this study performed a careful analysis of cooling protocols necessary to achieve vitrified organ storage, another important factor that will be critical to organ banking is the effect of the long-term storage temperature and conditions. The storage temperature should be sufficiently below $T_g$ to avoid the danger of devitrification, whereas storing at a temperature far below $T_g$ may also induce more brittleness inside the samples[25]. For instance, in one study, storage of VS55 for 6 months just below $T_g$ increased CWR from 50 to -100 °C/min[40], suggesting additional ice nucleation may have occurred during storage. For long-term storage (days to months to years) of human organ-scale vitrified samples, physical aging of the glass can also elevate $T_g$, decrease $T_d$ (devitrification temperature), and increase heat release at the glass transition during rewarming from the vitrified state[41,42]. This highlights the need for further study to identify the impacts of long-term vitrified storage and to identify optimal storage conditions. Further, CPA constituents (polymers, sugars, salts, etc.) can also impact stability in the brittle (i.e., glassy) state. For example, the thermal expansion coefficient of M22 (-2.52 × 10⁻⁴) is greater than VS55 (-1.84 × 10⁻⁴)[43], which could mean that M22 (which has polymers that may be more thermally expansive[44]) might be more prone to fractures from thermal stress.

We were able to demonstrate uniform volumetric rewarming using nanowarming in volumes up to 2 L. One hypothesized source of non-uniformity in larger RF coils is eddy current heating. However, prior analysis has assumed electrical properties at room temperature

when in fact, eddy current heating is expected to be insignificant at cryogenic temperatures, especially in the vitrified glassy state (see supplementary eddy heating section and Fig. S20).

While we have shown that nanowarming provides uniform and scalable heating across various volumes, larger organ systems will likely achieve the greatest benefits. Figure 9 (and Table S7) summarizes estimated nanowarming and convective rewarming rates for clinically relevant organs. While nanowarming offers substantial increases in rates and uniformity of heating for large organs, achievable convective rates may be higher (-2 times) for small, less vascularized organs (e.g., ovaries and testes). However, a lack of uniformity during convective rewarming may still lead to cracking in these cases.

The CPA concentration needed for physical vitrification at 0.5 L (relevant for human kidneys and hearts) is at least -8 M (40%EG + 0.6 Msucrose) and increases to -9.4 M (M22) at 3 L scale (relevant for human livers) as the minimum expected cooling rates are below 1 °C/min (Table S6). To achieve warming rates above the CWR of those CPAs, the required IONP concentration and the SAR of those IONP should be carefully assessed. For instance, Fig. 9 shows that for a human kidney, perfused IONP concentration of -10 mgFe/mL will provide a rewarming rate of -40 °C/min, which is safely above the CWR of M22. Figure 9 also shows that highly vascularized organs such as lungs (non-alveolar portions) and eyes will produce significantly higher nanowarming rates (-60 °C/min) than convection alone (see Table S7 for convective warming rates). Note that the predicted nanowarming rate for adult human kidneys is lower than previously reported rates achieved in rat kidneys (40 °C/min versus >60 °C/min)[15]. In this case, it is anticipated that the rates achieved in the small volume of rat kidneys included contributions from ambient warming and a higher IONP concentration in the surrounding CPA than was achieved solely due to the vascular fraction of the rat kidney.

Higher rewarming rates will also be possible at higher IONP concentrations or with higher heating nanoparticles. We heated 100 mg Fe/ml IONP in VMP and reached rates up to -1500 °C/min (Figs. S22 and 9C). Notably, consistent with earlier studies, it is also interesting that we find higher heating and $SAR_{Fe}$ in the cryogenic regime, likely due to the higher susceptibility and magnetization at such low temperatures and the reduced specific heat of CPAs in the glassy region[13]. It should be highlighted, too, that the above analysis is focused on the IONPs tested here and that other nanomaterials, such as nanowires/

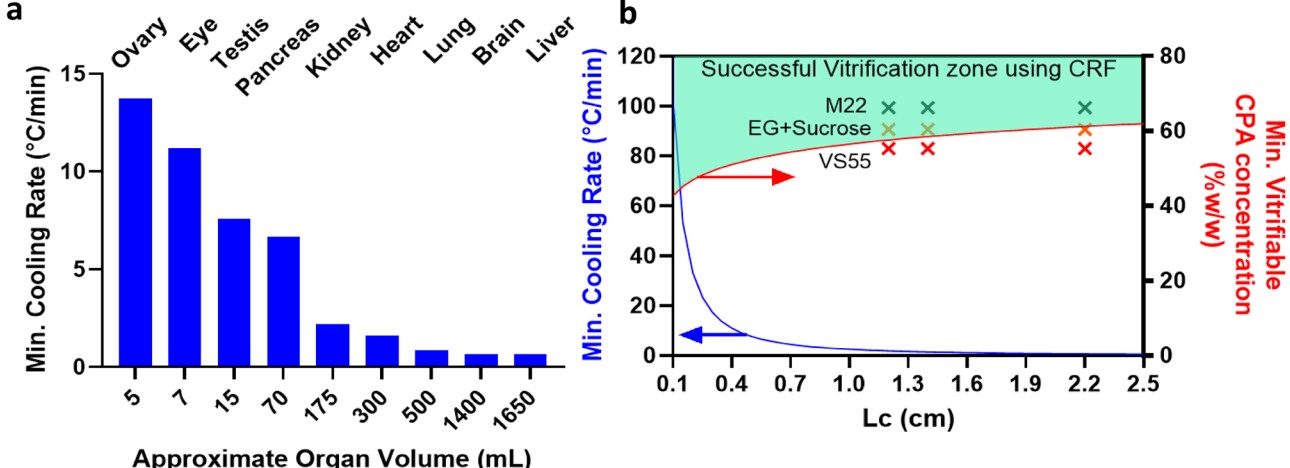

**Fig. 7 | Physical demonstration of liter scale nanowarming. a** Schematic showing 1 L (left) and 2 L (right) cryobag containing CPA with three fiber optic temperature probes (placed 4 cm apart / 13 cm depth inside 1 L and 15 cm depth inside 2 L cryobags). The dimensions of the cryobag are shown. **b** Temperature vs. time plot for rewarming of 1 L and 2 L volumes of M22 (differences in rates due to relative differences in IONP concentrations). Region with greatest risk of ice formation is shown as (*) and fracture failure as (**). Scatter plots of **c** average rewarming rate and **d** temperature difference ($\Delta T$) for 1 L and 2 L M22 (mean ± SD; $n = 3$ independent experiments from distinct samples). The rewarming rate is calculated from 0 to −100 °C. Average rewarming rates are greater than the CWR of M22 (-0.4 °C/min). Maximum temperature differences ($\Delta T$ between center and edges) are calculated between −120 to −150 °C (the glassy region).

**Fig. 8 | Achievable cooling rates and necessary CPA concentrations for human organ vitrification. a** Predicted minimum cooling rate (CR) as a function of volume of various human organs. **b** Plot of minimum cooling rate vs. $L_C$ (blue curve) and minimum vitrifiable CPA concentration in CRF as a function of $L_C$ (red curve). $L_C$ is calculated for various organs assuming ellipsoidal shape, and minimum cooing rate occurs at the center of geometry. $L_C$, CR, and dimensions of organs can be found in Table S6. Note that to predict cooling rates at different volumes, one should calculate $L_C$ and then use the blue plot in B. The experimental test points (x) showing success of M22, EG+Sucrose, and failure of VS55 at various $L_C$ (0.5, 1, 3 L volumes) are also plotted in B, validating the first-order approximation.

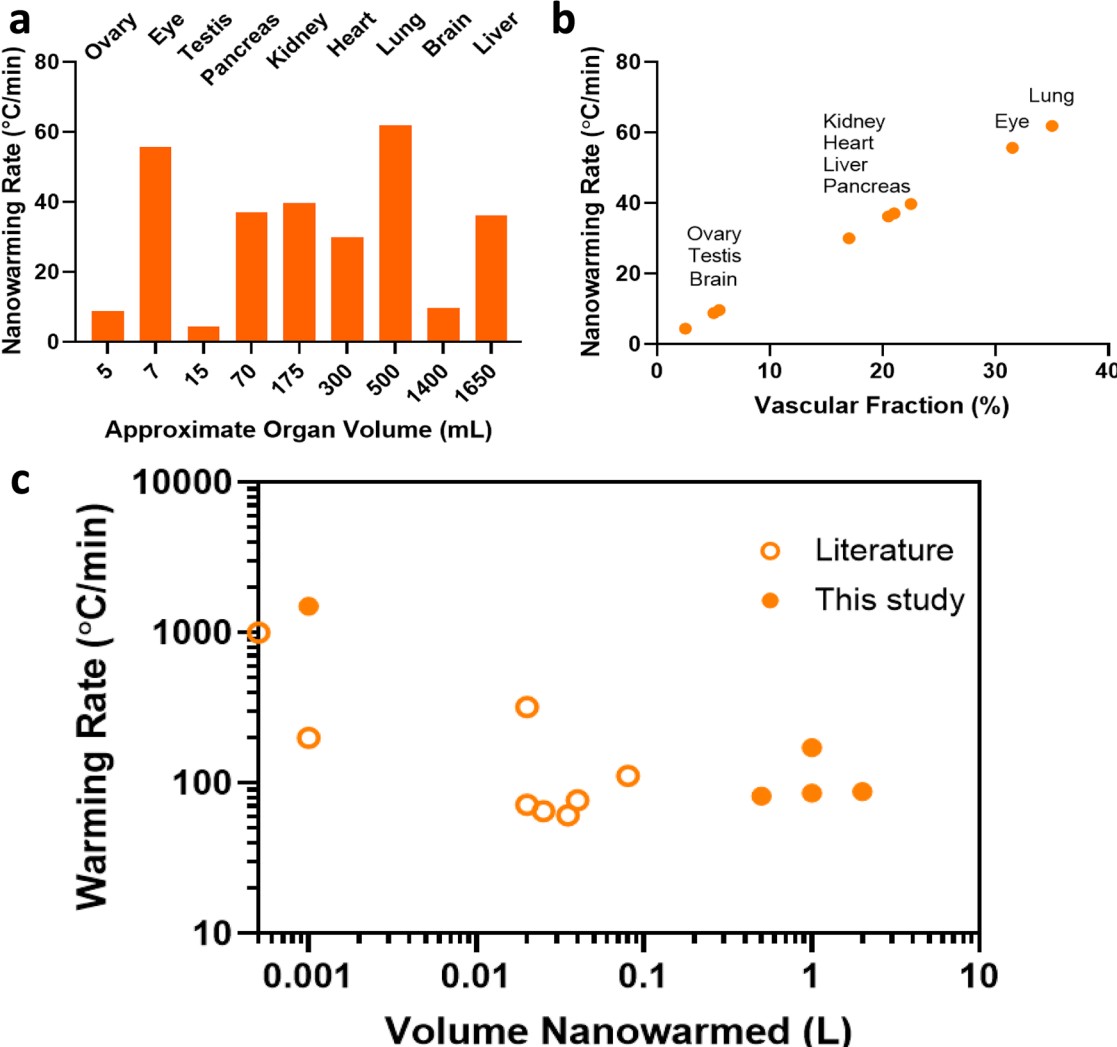

**Fig. 9 | Nanowarming rate predictions for human organs. a** Predicted nanowarming rate vs. total organ volume perfused at 10 mgFe/ml. **b** Predicted nanowarming rates vs. vascular volume of various organs perfused with 10 mgFe/ml. Rates are estimated for 10 mgFe/mL perfused IONP concentration with SAR$_{Fe}$-1050 W/gFe (see supplementary section for calculations and Table S7). **c** Nanowarming Rates for volumes from mL to L. Open circles represent literature values of warming rates achieved using nanowarming (see Table S8 for data values and references).

nanobars, have also shown ultra-rapid nanowarming heating rates up to ~1000 °C/min at much lower nanoparticle concentrations in proof of principle cryovial (<1 mL) samples (e.g., Cobalt Nickel nanowires in VS55)[45]. Still, these may be limited to external heating due to their aspect ratio, size, and concerns regarding biocompatibility.

Various other logistical factors must be considered to achieve human organ vitrification and nanowarming, including containers and IONP quantities. Verifying the cryogenic compatibility of the containers (i.e., polyimide, Teflon-PTFE, or other suitable material) and the ability to vacuum seal to remove the air interface as a nucleation site is key[46,47]. Further, we can estimate the IONP quantity requirements for nanowarming a whole human organ. Using a human kidney as an example, the amount of IONP can be estimated as ~0.4 gFe = 0.2 (vascular fraction) * 200 mL (organ volume) * 10 mgFe/mL (perfused IONP concentration). This amount should be multiplied by ~2 to 4 times the vascular volume to ensure complete perfusion loading[14,15], elevating the IONP required for a single kidney to ~0.8–1.6 gFe. IONP will also be needed for the surrounding solution. This highlights the need for scalable, biocompatible, colloidally stable IONP production[31].

In the future, several limitations of this work will need to be addressed. No biological assessment of the organ was performed, and further optimization and development of perfusion protocol will be required to minimize CPA toxicity and achieve consistent physical and biological success. For example, we assumed that CPA would fully equilibrate in tissue, but, as noted, tissue equilibration is often only in the range of ~92–94% after organ perfusion[15,34]. Nevertheless, experimental work on such dilutions of VMP and VS55 in tissue show similar vitrifiability to 100% CPA[48]. Furthermore, due to the limitations of the 120 kW RF coil diameter (Fig. S24), none of the vitrified livers in this study were nanowarmed. Future studies could also explore nanowarming of a smaller liver lobe in the existing coil. However, to achieve this one would require new protocols to successfully load and unload a liver lobe with CPA and iron-oxide nanoparticles, which was beyond the scope of this study. As this work is focused solely on assessing the feasibility for physical vitrification and nanowarming success (i.e., no biological and functional assessment), the toxicity of specific CPAs in specific organs will be addressed elsewhere[14,19,30,34,49]. Both physical and biological studies will be needed to ultimately achieve successful human scale organ banking.

## Methods
### Preparation of CPAs and Iron-oxide nanoparticle solutions
Three common vitrification CPAs were used in this study: M22, VS55, and 40%v/v EG + 0.6 M Sucrose, prepared according to their

formulations and composition as listed in Table S2. The preparation of the CPAs was completed by weight-volume percent (M22, VS55) and volume-volume percent (EG+sucrose) using a volumetric flask, as previously reported[14,19]. These CPAs were chosen due to their recent study in multiple organ systems, such as VS55 in rat hearts and kidneys[14,16–18], M22 in rabbit kidneys[34], and 40%v/v EG + 0.6 M Sucrose in rat livers[19]. Commercially available iron-oxide nanoparticles (IONPs) EMG308 (Ferrotec, Bedford, USA) (~10 nm) and silica-coated iron-oxide nanoparticles (sIONPs)[31] were used to prepare CPA + IONP mixture solutions. EMG308 was suspended in M22 with a modified carrier solution (water instead of LM5) to ensure the stability of these IONPs. sIONPs were synthesized as previously reported[31]. The colloidal stability of sIONPs and EMG308 in M22 was monitored by dynamic light scattering (DLS, see Fig. S1). The sIONPs and EMG308 were prepared in CPA (M22, VMP) or water. Recent work in organ vitrification was conducted with sIONPs[31,50], but uncoated EMG308 was used for simplicity in some physical demonstrations in these studies.

## Heat Transfer Finite Element Modeling (FEM)
Computational modeling was performed using commercial multiphysics simulation software (COMSOL 5.4). A 3D CAD geometry of a cryobag filled with CPA was created and simulated in COMSOL for heat transfer simulations (Fig. S2). Three cryobag volumes were simulated (0.5, 1 L, and 3 L) with dimensions close to experimental volumes (see supplementary on cryobag finite element modeling (FEM)). Details regarding governing equations, boundary conditions, initial conditions, and geometry are provided in the supplementary text and Table S4. All the experimental and modeled cooling and warming rates are reported in the 0 to −100 °C temperature range.

## Vitrification experiments
Three volumes, 0.5, 1, and 3 L, were evaluated for vitrification success. Polyethylene plastic "cryobags" of 2 mm thickness (McMaster Carr, Elmhurst, USA) were used to contain the CPA and organ samples (See Table S3 for bag sizes). Cryobags were heat-sealed at the top, and a clip was placed at the top to minimize air pockets and ice nucleation at the CPA-air interface. A controlled rate freezer (CRF) (Planar Kryo 560-II, Planar, Middlesex, UK) was used to execute cooling protocols for vitrification. Vitrified samples were stored in a −150 °C cryogenic mechanical freezer (MDF-C2156VANC-PA, Panasonic, IL, USA). Physical vitrification was verified by transparency observed by visual inspection and photography. Temperature measurements were conducted using fiber optic temperature probes and a four-channel monitoring system (Qualitrol, Fairport, NY, and Micronor Sensors, Ventura, CA, USA) at 1-2-second intervals using FO Temp Assistant Software or Optilink Software (provided by the manufacturer). The probes were pre-calibrated for any offset in the cryogenic temperature range. Probe placement inside the cryobag filled with CPAs is shown in Fig. S11.

## Porcine liver perfusion and vitrification
The University of Minnesota's Institutional Animal Care and Use Committee (IACUC) approved this study. Porcine livers ($n = 5$) were recovered from male, 9–30 kg, Yorkshire pigs ($n = 3$-25–30 kg, $n = 2$-9 kg) sourced from local vendors (Midwest Research Swine, Premium BioSource and MohrVet, MN). Animals were injected intravenously with 300 IU/kg heparin prior to euthanasia. After euthanasia, the abdomen was rapidly opened, the abdominal aorta and portal vein were cannulated, the thoracic aorta was cross-clamped, the suprahepatic vena cava (SHVC) was vented, and organs were flushed immediately with 5 L of a cold Histidine-Tryptophan-Ketoglutarate (HTK) solution. Once the flush was complete, the liver was explanted and placed in a cold HTK solution for transport. After back table preparation, the porcine liver was perfused via the portal vein with 40%EG + 0.6 M Sucrose in a step-loading protocol similar to that published for rat livers[19]. Specifically, 1xEC, 10%EG and 25%EG were perfused for 15 minutes each, followed by 40%EG + 0.6 M sucrose for 70 minutes at a constant flow rate of ~65 mL/min (see Fig. S5). More details can be found in the supplementary information.

## μCT Imaging for verification of vitrification
Microcomputed tomography (μCT) was used to verify physical vitrification (Fig. S13, S15). A modified foam/plastic container was used to hold the 0.5 L cryobags and porcine liver segments during μCT scanning, similar to that used previously for 20 mL systems[51]. The custom-made cooler can hold samples at cryogenic temperatures in a vitrified state (<$T_g$). The samples were scanned in a μCT imaging system NIKON XT H 225 (Nikon Metrology, MI, USA. Detailed information about μCT settings and image reconstruction can be found in the supplementary information.

## RF coil characterization (120 kW)
AMF LifeSystems, LLC. (Auburn Hills, MI) designed and built a custom-RF coil specifically for the development of nanowarming. Performance was characterized by measuring the spatial distribution across the full range of magnetic field values. To determine magnetic field strength (axial and radial component), we used a 2D high-frequency RF probe (AMF LifeSystems) placed inside the RF coils. We used an oscilloscope (LA354, LeCroy, NY) to record the voltage signals, which were then converted to magnetic field values. We used modeled as well as experimentally measured magnetic field distribution to compare spatial uniformity between the previously characterized 15 kW[6] and the new 120 kW RF coils (Fig. S16). The supplementary materials provide a detailed description of the RF coil in Fig. S16−S18 and Table S5.

## SAR Measurements
Specific absorption rates (SAR) measurements were performed on a 1 mL sample volume of CPA with iron-oxide nanoparticles (EMG-308, sIONPs at ~4 mgFe/mL) in a cryovial. The cryovial was placed inside the RF coils (15, 120 kW systems) within an insulated 3D-printed holder, and the temperature was recorded using fiber optic temperature probes. SAR was then calculated using the time rise method described elsewhere[26,52] (see supplementary SAR section, Fig. S19).

## Bulk nanowarming experiments
Liter-scale nanowarming was performed in the 120 kW RF coil described above. Volumes of 0.5, 1, and 2 L M22 with EMG308 were vitrified in heat-sealed cryobags with cylindrical shapes that fit within the RF coil (Fig. S24). Three fiber optic temperature probes were placed in the center, left, and right regions of the cryobag ~4 cm apart for all volumes using 3D printed jigs (Fig. S21). After vitrification, samples were stored overnight at −150 °C in a cryogenic freezer (MDF-C2156VANC-PA, Panasonic, IL). For rewarming, the sample was rapidly transferred from the freezer into the 120k kW RF coil (<20 seconds), placed in an insulated holder, and immediately rewarmed. The temperature was recorded every 1 sec during rewarming using the fiber optic thermometry system described above.

## Data analysis
All the graphs in the figures are plotted using GraphPad Prism 10. A total of ≥3 replicates are conducted for all the cooling, nanowarming, and SAR experiments. The number of replicates is listed in each figure legend. Replicates were performed on different days with distinct samples for livers and the same samples on different days for CPA experiments.

## Ethical declaration
This study complies with all relevant ethical considerations. The Institutional IACUC committee from the University of Minnesota (protocol #2111A39564) approved all animal studies.

## Reporting summary

Further information on research design is available in the Nature Portfolio Reporting Summary linked to this article.

## Data availability

All the data supporting this study's findings are available in the supplementary material. A source data file is also provided with this article. Any assistance in accessing or interpreting any data can be requested by contacting J.C.B. or E.B.F., corresponding authors (bischof@umn.edu or efinger@umn.edu) Source data are provided with this paper.

## Code availability

COMSOL 5.4 was used for computational heat transfer modeling and Altair Flux 2D for magnetic field modeling inside RF coils. The customized COMSOL codes are freely available from Zenodo (https://doi.org/10.5281/zenodo.16716098) and Flux 2D files are available from the corresponding authors upon reasonable request.

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

## Acknowledgements

This work was supported by the National Institutes of Health (NIH) Grant R01DK117425, NIH Grant R01HL135046, NIH Grant R01DK132211, and National Science Foundation (NSF) Grant EEC 1941543. We acknowledge support from a gift from the Biostasis Research Institute, funded partly through contributions from LifeGift, Nevada Donor Network, LifeSource, Donor Network West, and Lifebanc. The authors would like to thank the Dental CT laboratory for µCT scans of CPA volumes and the University of Minnesota, Minnesota Supercomputing Institute (MSI) for computational FEM access. We acknowledge Elliott Magnuson for the initial work on cryobags, Mikaela Hintz for technical assistance with CPA preparation, and Andy Grams for graphical design illustration. Lastly, we would also like to acknowledge Dr. Anirudh Sharma for his initial scientific work on RF coil scaling and Dr. Rhonda Franklin for scientific discussion on RF coil eddy heating.

## Author contributions

L.G. conceived and performed the study as a whole and contributed to all aspects with support and supervision from J.C.B. and E.B.F. L.G. carried out all experiments involving CPA, CPA+IONPs, porcine liver vitrification, and nanowarming with input from B.E.N., Z.H., and M.L.E. L.G. performed computational heat transfer modeling and SAR theoretical modeling with input from M.L.E. B.E.N. performed liver procurement surgeries and provided technical assistance in liver perfusion experiments. C.S. prepared all the CPA and IONP solutions and conducted stability studies. S.K. conducted relaxometry and ICP-OES for iron quantification. Z.H. performed µCT on all CPA samples with assistance from S.K. L.G. conducted all the SAR measurements, with some measurements performed by M.L.E. L.G. performed 120 kW RF coil characterization with assistance from R.G. R.G. performed the modeling of magnetic field distribution in RF coils. L.G. did all the data analysis. L.G. wrote the manuscript with input from M.L.E., E.B.F., and J.C.B. All authors reviewed, edited, and provided critical feedback. J.C.B. and E.B.F. acquired funding, and J.C.B. and E.B.F. oversaw project administration.

## Competing interests

M.L.E., E.B.F., and J.C.B. declare their interest in aspects of the technology described in these studies which have been filed in patents owned by the University of Minnesota. University of Minnesota has pending Patent Applications US14/775,998 and US17/579,369 (M.L.E. and J.C.B.), related to compositions and methods for rapid and uniform rewarming of cryopreserved biomaterials. R.G. has an interest in patents owned by AMF LifeSystems, LLC, and the 120 kW RF coil, which is their product. AMF Life Systems LLC has issued and pending Patent Applications US11877375B2, CA3034679C, CN109478797B, JP7246189B2, IL263861B2, EP3482476A4, and WO2018009542A1 (R.C.G), related to the RF coil used in this study. All other authors declare no competing interests.
