## [Transparent Peer Review file · Nature Communications]

Physical vitrification and nanowarming at liter-scale CPA volumes: Toward organ cryopreservation

Corresponding Author: Professor John Bischof

Version 0:

Reviewer comments:

Reviewer #1

(Remarks to the Author)

I found this paper very interesting, very easy to read, which in itself is quite a nice thing taking into account the topic could be very complicated. Congratulations to the authors for the clear effort presented in the manuscript.

I have some comments or suggestions that should be addressed prior publication:

In line 50 the authors state they developed the nanowarming technology, i suggest to add a reference where the basics of this technology are presented to provide context for the readers that might be encountering this topic for the first time. Looking at figure 1, it looks like vitrification only happens in the bag below and not in the one with IONPs, i suggest a minor modification there.

I consider that the statement "This highlights the importance of 171 appropriately designing cooling protocols for a given geometry" in line 172-73 needs to be further explained as it plays a crucial part on the applications moving forward. Maybe further discussed in the final section.

Will the organs be all surrounded in vitrification solution when vitrified? How much volume will they be vitrified in? Do the authors have any estimates of the organ/liquid volume ratio needed for different organs? If so I consider the discussion could be enriched by this content being added.

Reviewer #2

(Remarks to the Author)

This is an interesting and important work on successful vitrification of large (liter)-scale cryoprotectant solutions and porcine liver and further successful warming of the cryoprotectant solutions. The authors conducted both extensive modeling and experimental analyses of the large-volume vitrification technology. This work may facilitate successful human organ banking in the future. Below are a few comments that should be carefully addressed:

1. The title is misleading: "human organ scale" should be changed to "liter scale" because no human organ is studied in this work. Although it is good to make some connection to human organs in terms of volume in the Abstract and Introduction, it is inappropriate to include human organ in the title.
2. Similarly, the inclusion of human organs in Fig. 1 is misleading, because it shows as if human organs are successfully vitrified and rewarmed by nanowarming in this work (while they are not). The human organs should be removed from this figure.
3. The modeling results in Fig. 2b for the first 90 min when it is critical for vitrification, are quite off compared to the experimental data at the center. What are the possible causes of it? Without rigorous validation of this model, any prediction (including the data shown in Fig. 9 and several supplementary figures) with the model would be unconvincing.
4. How many times were the experiment for Fig. 4 done independently? It is desired to show such images from at least 3 independent experiments to demonstrate repeatability. The additional four images can be added as a supplementary figure.
5. Did the authors successfully warm the apparently vitrified liver and check the tissue viability and integrity? These data would be of super interest. This should be made clear in the Abstract, as well.

Reviewer #3

(Remarks to the Author)

The authors tested the ability of three cryoprotective agent solution to vitrify and rewarm at large volume, and vitrified a porcine liver as proof of principle. There are several issues to be clarified:

Major concerns:

- 1.The successful vitrification process, as demonstrated in the present study, depends upon the physical properties of the cryopreservation solution. This process was achieved by modifying the formulation of the CPA. For instance, the M22 (~9.4M) formulation utilized in this research was attained by increasing the concentration of the CPA (eg. 40%EG+0.6M sucrose is ~8M), which resulted in a lower critical cooling rate (CCR) and ultimately facilitated vitrification. It is well-known in cryopreservation field, the authors also listed several references in SI Table S2. However, a critical issue should be addressed, from the clinical viewpoint, the biosafety concerns associated with high-concentration solutions are very important, basic biological characterization and function of the organ after cryopreservation are necessary to provide to prove the biosafety.
- 2.In addition, the thermal stress of biological samples may differ significantly from that of cryoprotect solution alone. Organs, which comprise fibrous structures and blood vessels, exhibit significantly more complex coefficients of thermal expansion and moduli compared to solutions. Therefore, the analysis of thermal stress should not depend on simplistic models that are only applicable to solutions.
- 3.Fig. 2d displays clearly the presence of white ice crystals formation on its upper section, it seems the result could not support the description in line 151-152 "Photos of a successful vitrified (glass) M22 inside a cryobag for ...d 3 Liter (largest volume reported)."
- 4.Fig.2: In addition, the M22-0.5L cryobag with cryoprotectant exhibits good transparency, and the white letters behind the cryobag can be displayed. These phenomena did not occur in the M22-3L groups. The transparency of these two groups appears to be different, which may reflect the different states of solutions. A similar phenomenon also occurred in the 40% EG+0.6M sucrose group. According to Fig S10, the letters behind the 0.5L and 1L groups can be clearly seen, while the letters cannot be seen in 3L group. And the authors claim that "40% EG+0.6M sucrose successfully vitrified at 0.5 and 1L but not 3L." Why M22-3L group and 40% EG+0.6M sucrose-3L group have similar phenomenon?
- 5.Another question is that why there is an absence of frosting on the samples when they were removed from the cryogenic refrigerator?
- 6.Table 1 and Fig. S4, the authors compare the cooling procedure with different starting temperatures (-20 °C, 0 °C, 20 °C), it is better to explain the reason for choosing 0 °C as the starting temperature for cooling.
- 7.Fig.S13: The presence of vitrification images of cryobag with cryoprotectant alone can not represent vitrification condition of organ in the cryobag with cryoprotectant. It is essential to provide more explicit and direct evidence, such as micro computed tomography (microCT) characterization, to show the vitrification outcomes of organ (especially the internal state of organ), not only surface optical photograph (Fig.4). In addition, CT characterization of the liver with cryoprotectant loading EMG308 particles should also be provided to show the vitrification condition.
- 8.SI, line 157: "perfused CPA long enough to achieve >96% equilibration in tissue to demonstrate physical vitrification success (verified using the effluent reflective index)." Please provide more evidence to support this statement.

Minor concerns:

- 1.Considering that figures serves as the primary illustration in the main text, the image quality of Fig.2 can be further improved.
- 2.Fig.3: The picture is not clear, the shaded part of Fig.3 b,c affects the clarity.

Version 1:

Reviewer comments:

Reviewer #1

(Remarks to the Author)

Reviewer #2

(Remarks to the Author)

Thank the authors for carefully addressing my comments. I still feel the title of the manuscript is misleading and the last four words "to enable organ cryopreservation" in the title should be replaced with "of cryopreservation solutions", as the authors clarified that they did not warm the vitrified porcine liver in this work. Other than this, the work is acceptable for publication in this journal.

Reviewer #3

(Remarks to the Author)

The revised manuscript demonstrates progress through the addition of thermal simulations and repeated liver vitrification experiments. However, several critical issues require clarification to strengthen the scientific rigor and resolve reader confusion.

Specific Comments:

1. Title-Abstract Discrepancy & Experimental Scope Clarification

The title claims "Physical vitrification and nanowarming at liter volume to enable organ cryopreservation", yet the abstract explicitly states "...we successfully vitrified although did not rewarm...". This inconsistency creates confusion regarding the study's achievements. While the authors successfully demonstrated vitrification and nanowarming in CPA solutions (1L to 3L), organ (226ml to 752ml) cryopreservation remains incomplete due to the absence of organ rewarming (a complete cryopreservation includes cooling process and warming process).

The justification for not rewarming organs - "the size of RF coil is smaller than organ dimensions" - conflicts with experimental evidence:

- The 3L CPA group (cryobag dimensions: 19×30×10.5 cm) underwent successful nanowarming, yet the 0.752L organ (max dimension ~20 cm) was deemed incompatible with the RF coil (13.3 cm diameter limit).
- Similarly, the 1L CPA cryobag (19.5×14.5×6.5 cm) accommodated nanowarming, while a 0.752L organ (smaller than 1L cryobag dimensions) could not.

This discrepancy requires clarification: Why can nanowarming be achieved in larger-volume CPA systems but not in smaller-volume organs? A quantitative comparison of spatial constraints (e.g., organ geometry vs. RF coil configuration) should be provided.

2. Inaccurate Organ Volume Description

The manuscript describes porcine livers as "~0.6-1L total volume", yet the actual experimental volumes (Fig. S14) range from 0.226L to 0.752L (the actual volume of porcine livers used in the experiment is 750 ml, 650 ml, 752 ml, 226ml and 250 ml, respectively). The "~0.6-1L" claim is misleading and should be revised to reflect the true data range (0.226-0.752L). If smaller livers (<0.6L) were intentionally excluded from analysis, this must be explicitly stated with justification.

3. Figure 4 Caption & Annotation Issues

- Missing labels: Captions for Fig. 4c and 4f are absent.
- Ambiguous annotations:
 - The statement "Blue (*) indicates perihilar tissue, fatty tissue, and the cryobag" is confusing. The cryobag appears to be represented by bubbles between the organ and bag wall; a distinct symbol (e.g., †) should be used for clarity.
 - The description of "surface ice condensation (**)" lacks spatial context. The bisected liver region should be clearly marked in Figs. 4a,b,d,e. And the photography taking also need to maintain low-temperature condition to avoid extra ice formation.

4. Optimization of the CPA perfusion loading protocol

Ice formation occurred during vitrification in organ sample 2 and 3 (Fig.S14), the author explained that "further optimization of the CPA perfusion loading protocol may be required (which was not the focus of this manuscript)". Since CPA perfusion and distribution in organ can directly affect vitrification and rewarming outcomes, and it is also a physical problem. Optimization of the CPA perfusion loading protocol is suggested, or the authors could prove that a trace amount of ice at the periphery will not affect the function of the organ by biological data.

5. Biological Validation Deficiency

While Reviewer #2 and #3 previously requested basic biological assessments (tissue viability, integrity), the authors only provide visual inspection of 100-200mL segments (Fig. S15). Given the RF coil limitation for whole-organ rewarming, minimum validation could include:

- Cell viability assays (e.g., live/dead staining) on rewarmed tissue segments
- Structural integrity analysis (H&E staining) to assess CPA toxicity and ice damage
- Functional metrics (e.g., ATP levels) for sub-samples

Even preliminary biological data would significantly strengthen the claim of "feasibility for organ cryopreservation", as physical vitrification alone cannot confirm biospecimen functionality.

Additional Suggestions:

- Consider renaming the title to "Physical vitrification and nanowarming at liter-scale CPA volumes: Toward organ cryopreservation" to better align with demonstrated results.
- Include a schematic of RF coil dimensions vs. organ/cryobag geometries in supplemental materials.

This revision can enhance clarity while preserving the original intent.

Version 2:

Reviewer comments:

Reviewer #3

(Remarks to the Author)

After thorough evaluation of the authors' responses and revised manuscript, I maintain significant concerns regarding the scientific validity and novelty of this work. My objections are outlined below:

1. The authors repeatedly emphasize this as a "physical proof-of-concept" study, asserting that biological validation "exceeds the scope." Cryopreservation research ultimately aims to preserve "biological function". Physical vitrification/rewarming without any viability assessment (e.g., cell survival, structural integrity, or functional recovery) fails to

demonstrate feasibility for organ cryopreservation.

2.As to the physical approach, there is a lack of methodological and material innovation. The study offers no significant advances over existing literature, nanowarming via AMF is well-established (e.g., Manuchehrabadi et al., *Sci Transl Med*, 2017; Sharma et al., *Adv Sci*, 2021). M22 CPA is not novel (Fahy et al., *Cryobiology*, 2004), and the authors implemented it without modifications to enhance biocompatibility or efficacy. The core work-screening CPA compatibility for large volume constitutes incremental engineering optimization, which is insufficient. If the focus is purely physical, this work belongs in specialized journals.

3.Only 1 of 3 porcine livers (752 mL) achieved full vitrification, others (650 mL, 720 mL) showed ice formation (Fig. S14). The authors attribute this to "biological variability," but no data supports this claim. This inconsistency undermines the physical protocol's robustness.

4.The authors claim rewarming organs was impossible due to RF coil constraints (diameter: 13.3 cm). However, it seems that ~200 mL liver could fit Cryobag #2 (~11×13×30 cm; Fig. S24), which was used for rewarming experiment. Rewarming tissue segments (e.g., 100–200 mL) to assess viability was technically feasible yet omitted.

5.The response letter and manuscript contain grammatical errors (e.g., "Different cryobags than those used..." → should be "...that..."), reflecting insufficient revision rigor.

Point by point response to review of **NCOMMS-24-71333**

Dear Reviewers,

We thank you for your careful read and helpful comments regarding our work. In response, we have revised the manuscript and supplemental material incorporating the suggestions and answering the questions posed. We have included additional replicates of pig liver vitrification, modified figures to make their message more clear, and show additional results confirming our ability to vitrify at liter scale. This includes micro-CT imaging showing successful vitrification of pig livers. Below we address each of the comments and questions raised by the reviewers and editors in blue text.

REVIEWER COMMENTS

Reviewer #1 (Remarks to the Author):

Comment #1: I found this paper very interesting, very easy to read, which in itself is quite a nice thing taking into account the topic could be very complicated. Congratulations to the authors for the clear effort presented in the manuscript.

Authors would like to thank the reviewer for their comments. It is encouraging to know that they found our contribution to be both interesting and easily comprehensible.

Comment #2: I have some comments or suggestions that should be addressed prior publication: In line 50 the authors state they developed the nanowarming technology, I suggest to add a reference where the basics of this technology are presented to provide context for the readers that might be encountering this topic for the first time.

Response: In line 50, we have added 2 references which are the initial papers describing the development of the nanowarming technology.

Added material:

"Etheridge, Michael L., et al. "RF heating of magnetic nanoparticles improves the thawing of cryopreserved biomaterials." Technology 2.03 (2014): 229-242.

Manuchehrabadi, Navid, et al. "Improved tissue cryopreservation using inductive heating of magnetic nanoparticles." Science translational medicine 9.379 (2017): eaah4586."

Comment #3: Looking at figure 1, it looks like vitrification only happens in the bag below and not in the one with IONPs, I suggest a minor modification there.

Thank you for noting that this was not clear in the figure. Indeed, both the CPA alone and the CPA+IONP cryobags vitrify. We have modified Fig.1 as below to make it clearer that both were vitrified.

Revised Figure 1:

Fig. 1: Schematic flow of steps (left to right) in liter scale vitrification and rewarming. Liter volumes of a CPA (0.5-3L) in cryobags are large enough to hold a human organ. The cryobag is placed inside a controlled rate freezer (CRF) for cooling. For nanowarming (top section of the flow chart), the cryobag is vitrified with iron oxide nanoparticles (IONPs) suspended in the CPA. The vitrified cryobag is placed inside the RF coil, alternating magnetic fields are turned on, and the IONPs generate heat. This leads to successful (rapid, uniform) rewarming, avoiding crystallization or cracking failure modes even at the liter scale. Traditional rewarming employs convection using a water bath, which at these scales results in ice recrystallization and/or fractures due to insufficient rewarming rates (slow) and thermal stresses (non-uniform), respectively (lower section of the flow chart). IONP, iron oxide nanoparticles; CPA, cryoprotective agent; CRF, controlled rate freezer.

Comment #4: I consider that the statement "This highlights the importance of appropriately designing cooling protocols for a given geometry" in line 172-73 needs to be further explained as it plays a crucial part on the applications moving forward. Maybe further discussed in the final section.

Thank you for noting that this was not clear in the text. We have added more information after "This highlights... a given geometry" as noted below to further describe the importance of the geometry of the system in designing cooling protocols. Additionally, we have added reference(s)

to Supplementary Figures S4 & S7 and section “Porcine Heat Transfer Modeling” which discusses the key factors to consider while designing a cooling protocol based upon the characteristic length, L_c , of a given geometry.

Added material:

“This highlights the importance of appropriately designing cooling protocols for a given geometry. For instance, different shapes and form factors can be attained for a given volume resulting in different characteristic lengths, L_c . This difference in L_c will change the convective cooling process for the system in the CRF, such as the cooling rate above T_g , annealing time, slow cooling rate in glassy region, and storage hold time (see Figure S4 and S7 and supplementary section on porcine heat transfer modeling).”

Comment #5: Will the organs be all surrounded in vitrification solution when vitrified? How much volume will they be vitrified in? Do the authors have any estimates of the organ/liquid volume ratio needed for different organs? If so I consider the discussion could be enriched by this content being added.

Theoretically organs do not need a surrounding volume for vitrification as the extra volume becomes an additional thermal mass which slows the cooling rate. Table S6 presents this limiting case. However, there are several good reasons to include CPA surrounding organs during vitrification which we adopted during porcine liver vitrification. First, the CPA helps prevent any cryobag surface induced ice nucleation during cooling due to presence of heterogenous impurities. Second, the CPA helps to eliminate air gaps which act as an insulating layer thereby impeding heat transfer and hence slowing cooling rates. Thirdly, CPA can act as a protective layer during transfer. Lastly, the surrounding CPA volume when augmented with IONPs will provide supplemental heating during nanowarming. To approximate the amount necessary for these surrounding volumes, we have provided a simple analysis and added it in the supplementary section entitled “Estimation of surrounding CPA volume for human organs during cooling.” We have also added a Figure S23 showing the geometry to accompany these calculations.

Added material:

Estimation of surrounding CPA volume for human organs during cooling

Here we provide an example calculation for CPA volumes to be added to a human organ during nanowarming cryopreservation assuming ellipsoidal bag geometry. Note this is meant only as an example rather than a universal solution as the calculation depends upon organ shape, sterility, cryobag form factor, container requirements, cooling technology, rewarming technology and transport logistical constraints which can vary between users.

For our example we assume an organ is laid horizontally (flat) with its smallest dimension i.e. thickness as the vertical height, as shown in Figure S23. To avoid risk of ice nucleation due to air at the cryobag surface, the organ is assumed to be completely submerged in CPA. This would imply the smallest dimension, i.e. thickness, would be the height of surrounding volume around each organ.

$$\begin{aligned} \text{Total volume (Surrounding solution + organ volume) (TV)} \\ = \pi * \frac{\text{width}}{2} * \frac{\text{thickness}}{2} * \text{length} \end{aligned}$$

$$\text{Volume of an organ (V)} = \frac{4\pi}{3} * \frac{\text{length}}{2} * \frac{\text{width}}{2} * \frac{\text{thickness}}{2}$$

$$\frac{TV}{V} = \frac{3}{2} = 1.5$$

$$SV = TV - V = \frac{\pi}{3} * \text{length} * \frac{\text{width}}{2} * \frac{\text{thickness}}{2}$$

$$\frac{SV}{V} = 0.5$$

Figure S23: Schematic of ellipsoidal organ and cryobag geometry for calculations of human organ Lc and surrounding volumes.

Even though this simple ellipsoidal calculation results in a constant surrounding CPA volume to total volume ratio, in real life scenarios this surrounding CPA volume should be designed and adjusted according to the specific use case.

Reviewer #2 (Remarks to the Author):

Comment: This is an interesting and important work on successful vitrification of large (liter)-scale cryoprotectant solutions and porcine liver and further successful warming of the cryoprotectant solutions. The authors conducted both extensive modeling and experimental analyses of the large-volume vitrification technology. This work may facilitate successful human organ banking in the future.

The authors thank the reviewer for their positive comments about the importance of this work.

Comment #1. The title is misleading: “human organ scale” should be changed to “liter scale” because no human organ is studied in this work. Although it is good to make some connection to human organs in terms of volume in the Abstract and Introduction, it is inappropriate to include human organ in the title.

The authors thank the reviewer 2 for their comment we agree that the original title might be misinterpreted. We have removed “human” from the title and suggest the following alternative:

“Physical vitrification and nanowarming at liter volumes to enable organ cryopreservation”

Comment # 2: Similarly, the inclusion of human organs in Fig. 1 is misleading, because it shows as if human organs are successfully vitrified and rewarmed by nanowarming in this work (while they are not). The human organs should be removed from this figure.

We believe including human organs in the schematic in Figure 1 is important to conceptually describe the intended end use and potential impact of the work. However, we have modified Figure 1 to indicate that the organ use is a “future use”.

Comment 3. The modeling results in Fig. 2b for the first 90 min when it is critical for vitrification, are quite off compared to the experimental data at the center. What are the possible causes of it? Without rigorous validation of this model, any prediction (including the data shown in Fig. 9 and several supplementary figures) with the model would be unconvincing.

Modeling was used to bracket the experimental behavior. More specifically, an exact agreement is not expected due to uncertainty in the exact thermal probe placement (Figure S3 and S11) with the temperature probe jig (Figure S11) and due to cryobag deformation (i.e. thermal contraction) during cooling (Figure S11). The modeled behavior would then be expected to bracket the observed experimental measurements, with the modeling results presenting the full spatial distribution within the cryobag. We were thus able to predict vitrification, minimum and maximum rates, thermal equilibration (annealing times), and ΔT_{\max} in the sample within a reasonable range of agreement with our experiments. For instance, cooling rates within 0 to -100 °C (Figure 3d), equilibration within $\pm 2^\circ\text{C}$ during annealing, and temperature difference (ΔT_{\max}) (Figure 3e) shows agreement between modeling and experiments. In addition, this model (although with a modified geometry) was first introduced in [1] and then compared directly to organ cooling experiments in [2] with good agreement. This, along with our ability to

predict vitrification success and failure in all of our large CPA volumes suggests the model is sufficient for the presented purpose.

Comment #4. How many times were the experiment for Fig. 4 done independently? It is desired to show such images from at least 3 independent experiments to demonstrate repeatability. The additional four images can be added as a supplementary figure.

Initially, we conducted 2 trials of vitrification of porcine livers. The largest of these was shown in main manuscript results (now modified Figure 4). We now include 3 additional livers (n = 5 in total) and have included images and summary table of each of these in the supplementary data (Fig. S14) and modified method and results section to incorporate the repeats as suggested. Each of the livers were successfully vitrified, but some small areas of ice were noted in the two largest livers (650 and 752 gm), mostly in the less well vascularized periportal adipose tissue and some on the surface, as shown below. These results indicate that physical vitrification of livers is possible, however, as noted in the manuscript, further optimization of the CPA perfusion loading protocol may be required (which was not the focus of this manuscript). Also, just to note, the final two runs were done with smaller livers due to limitations in the availability of animals during the response window.

Added material:

Liver No.	~Liver Size (gms)	~Total Volume (mL)	Cryobag Size	Results
1	750	950	25.4 x 20.3 cm (Lc ~ 0.9 cm)	Fig. 4
2	650	900		Fig. S14
3	752	1000		
4	226	630	20 X 20.3 cm	Fig. S14
5	250	650	(Lc ~ 0.7 cm)	

Fig. S14: Photos of vitrified porcine livers from this study. A total of n=5 livers were vitrified as described in the Table. Four livers are shown here and the final liver (Liver 1) in main text (Fig. 4). Note in general the transparent vitrified CPA surrounding the vitrified liver tissue. There were some cases of surface ice on the perihilar fatty tissue around the portal vein and vasculature, likely due to insufficient CPA penetration in this less well vascularized tissue. Furthermore, the largest livers, 1 and 3, showed a trace amount of ice at the periphery, although the bulk of the liver vitrified. This indicates that physical vitrification of livers is possible, but that further development of the CPA perfusion loading protocols is required.

5. Did the authors successfully warm the apparently vitrified liver and check the tissue viability and integrity? These data would be of super interest. This should be made clear in the Abstract, as well.

We did not rewarm nor test the viability of the liver in this study. The focus of this study is on demonstrating the feasibility of physically vitrifying and rewarming at the liter scale. We have made this clearer in the modified abstract as shown below. Further study on biological outcomes is certainly an area of future study and focus, but beyond the scope of the current manuscript.

Added material:

“As additional proof of principle, we successfully vitrified although did not rewarm, several porcine livers (0.6-1L total volume).”

Our intention with this work was to find physical limitations for both vitrification and nanowarming. With regards to nanowarming, we demonstrated rewarming in the largest volume that could be accommodated in our RF coil. We vitrified the porcine livers (~1L volume, >20cm largest dimension) lying flat, which would not fit in the current geometry of our 120 kW RF coil which will only accommodate ~13.3 cm diameter of a cylindrical volume (Table S5). However, the only limitation is on the design of the current RF coil and the physical principles of scaling indicate that rewarming a vitrified liver is possible in future study.

Reviewer #3 (Remarks to the Author):

The authors tested the ability of three cryoprotective agent solution to vitrify and rewarm at large volume, and vitrified a porcine liver as proof of principle. There are several issues to be clarified:

Authors would like to thank reviewer 3 for their comments and feedback, all of which enhances the readability, clarity and technical soundness of this manuscript.

Major concerns:

1. The successful vitrification process, as demonstrated in the present study, depends upon the physical properties of the cryopreservation solution. This process was achieved by modifying the formulation of the CPA. For instance, the M22 (~9.4M) formulation utilized in this research was attained by increasing the concentration of the CPA (eg. 40%EG+0.6M sucrose is ~8M), which resulted in a lower critical cooling rate (CCR) and ultimately facilitated vitrification. It is well-known in cryopreservation field, the authors also listed several references in SI Table S2. However, a critical issue should be addressed, from the clinical viewpoint, the biosafety concerns associated with high-concentration solutions are very important, basic biological characterization and function of the organ after cryopreservation are necessary to provide to prove the biosafety.

The authors agree that the biosafety of a cryoprotectant is crucial for cryopreserved organ recovery and function. We appreciate first-hand the complexity of studying organ and tissue specific injury due to cryoprotectant toxicity and osmotic damage. However, the current manuscript is concerned with the physical aspects of vitrification and rewarming. Demonstrating that appropriately designed CPAs can be successfully vitrified and rewarmed at the liter scale motivates and informs continued biological study of appropriate CPAs for use in organ vitrification. Further, the choice of CPAs studied was motivated by prior study on CPAs with promising (although not perfect) toxicity profiles, such as M22 [3-5], 40%EG+0.6M Sucrose [6, 7] and VS55 [2, 8-9]. This limitation is now more clearly addressed in the revised title and revised discussion section as shown below.

Added material:

“As this work is focused solely on physical vitrification success, the toxicity of specific CPAs in specific organs will be addressed elsewhere. Both physical and biological studies will be needed to ultimately achieve successful organ banking”.

2. In addition, the thermal stress of biological samples may differ significantly from that of cryoprotect solution alone. Organs, which comprise fibrous structures and blood vessels, exhibit significantly more complex coefficients of thermal expansion and moduli compared to solutions. Therefore, the analysis of thermal stress should not depend on simplistic models that are only applicable to solutions.

In this study, we developed cooling protocols to minimize temperature non-uniformity in the glassy phase of the vitrified sample and avoid thermal stress induced cracks. Thermal expansion coefficients for CPA-permeated liver tissue, has not been studied in literature, so we elected to use pure CPA solution values instead as an approximation. We believe this is

appropriate as available literature on arteries and muscle tissue suggests that CPA thermal expansion coefficient exceeds that of CPA permeated tissue coefficients [10-12]. This suggests that our assumption of CPA coefficients will provide a conservative lower limit of ΔT_{max} for predicting fracture failure in the organ [1]. Using these CPA coefficients, we calculate (using the thermal shock equation) that fracture is not predicted for $\Delta T_{max} < 20^{\circ}C$ in Figure 2, whereas fracture is predicted above this, as shown in Figure S9. Our past studies in rat kidneys [2, 5], hearts [9], and livers [7] have also shown that by maintaining temperature differences within $\Delta T_{max} \leq 20^{\circ}C$, fractures can be avoided in practice. Therefore, while we agree that literature would benefit from a more comprehensive study of the thermomechanics of vitrified tissue, we believe that our study's focus on motivating physical behavior (i.e. thermal gradients, which lead to thermal stress) provides a compelling description of the principles needed to successfully vitrify liter scale volumes.

3. Fig. 2d displays clearly the presence of white ice crystals formation on its upper section, it seems the result could not support the description in line 151-152 "Photos of a successful vitrified (glass) M22 inside a cryobag for ...d 3 Liter (largest volume reported)."

We have modified Figure 2d to provide a better picture of the 3L vitrified cryobag. Furthermore, upon close inspection, there is no ice seen in the upper section.

Revised Figure 2:

Fig. 2: Demonstration of physical success of vitrification in multiple volumes. a Table summarizing vitrification results for all the three CPAs and volumes. Photos of successfully vitrified (glass) M22 inside a cryobag for b 0.5 Liter, c 1 Liter, and d 3 Liter

(largest volume reported). The out-of-plane thicknesses are 5.5, 6.5, and 10.5 cm for 0.5, 1, and 3L cryobags, respectively.

4. Fig.2: In addition, the M22-0.5L cryobag with cryoprotectant exhibits good transparency, and the white letters behind the cryobag can be displayed. These phenomena did not occur in the M22-3L groups. The transparency of these two groups appears to be different, which may reflect the different states of solutions. A similar phenomenon also occurred in the 40% EG+0.6M sucrose group. According to Fig S10, the letters behind the 0.5L and 1L groups can be clearly seen, while the letters cannot be seen in 3L group. And the authors claim that “40% EG+0.6M sucrose successfully vitrified at 0.5 and 1L but not 3L.” Why M22-3L group and 40% EG+0.6M sucrose-3L group have similar phenomenon?

It should be noted that a larger cryobag volume results in a larger thickness of sample geometry and thereby also alters the refraction of light waves (distortion) coming out from the white letters in the background, such that it is more difficult to clearly see through it. Also, based on the reviewer comments, we noted that the 3L M22 cryobag seems not to have been wiped down (with methanol) as effectively as the other volumes. This resulted in some condensation and poorer visibility in the image. We have now updated 3L M22 with a better photo in Fig. 2 (and Fig. S10) where background CRF letters are more visible.

Further, in Figure S10, for 3L 40%EG+0.6MSucrose, the white letters in the background are not clearly visible because 3L volume forms ice in the center, which is evident as white spheres (black arrows). Note, upon looking at the side photos (**shown below – no caption provided**) of 3L M22 vs. 40%EG+0.6M Sucrose, it is easier to distinguish ice formation in the latter.

Revised Figure 2:

Fig. 2: Demonstration of physical success of vitrification in multiple volumes. a Table summarizes vitrification results for all the three CPAs and volumes. Photos of a successful vitrified (glass) M22 inside a cryobag for b 0.5 Liter, c 1 Liter, and d 3 Liter (largest volume reported). The out-of-plane thicknesses are 5.5, 6.5, and 10.5 cm for 0.5, 1, and 3L cryobags, respectively.

5. Another question is that why there is an absence of frosting on the samples when they were removed from the cryogenic refrigerator?

The vitrified samples were quickly cleaned and wiped using methanol to reduce frost condensation during photography. This technique has been previously used in literature [13].

6. Table 1 and Fig. S4, the authors compare the cooling procedure with different starting temperatures (-20 °C, 0 °C, 20 °C), it is better to explain the reason for choosing 0 °C as the starting temperature for cooling.

In principle, one can choose any starting CRF temperature below the perfusion temperature and above T_g . However, here we show that the impact on cooling rates is minimal for larger volumes, as shown in Figure S4. We now explicitly mention the reason for different start temperature selections in the supplementary computational modeling section on Page 5.

Added Material:

“However, the CRF start temperature should not be higher than the organ perfusion temperature to minimize the potential for increased CPA toxicity. For this reason, we chose 0°C, which is slightly below our perfusion temperature (~4°C).”

7. Fig.S13: The presence of vitrification images of cryobag with cryoprotectant alone cannot represent vitrification condition of organ in the cryobag with cryoprotectant. It is essential to provide more explicit and direct evidence, such as micro computed tomography (microCT) characterization, to show the vitrification outcomes of organ (especially the internal state of organ), not only surface optical photograph (Fig.4). In addition, CT characterization of the liver with cryoprotectant loading EMG308 particles should also be provided to show the vitrification condition.

We used our previous μ CT protocol developed for imaging vitrified rat organs to confirm vitrification in 0.5L volumes (Figure S13). Since larger porcine livers (~1L) exceed the capacity of the previously used μ CT device (NIKON XT H 225), we initially elected to only show cross-sectional, visual photos as evidence of vitrification (Figure S14) in addition to the surface photography in Figure 4.

However, now we have added Figure S15, which shows μ CT images of segmented, vitrified sections of pig liver. Note that we have sample volume limitations for our μ CT machine, and therefore we were only able to scan smaller segments (<500mL) of vitrified liver. For comparison, we have also added a μ CT image of a frozen liver section. Vitrified liver regions have higher radiodensity (Hounsfield Units (HU) >600). In contrast, frozen regions have lower HU (<500), as is expected from higher density vitrified vs. frozen tissue as we previously demonstrated in rat livers [7]. Additionally, we added visual photos of sectioned, vitrified liver alongside both the front and back view of whole intact vitrified liver in main Figure 4.

Our past studies for rat organs (liver, kidney, and heart) demonstrated that IONP+CPA-loaded organs vitrified similarly to CPA-loaded organs alone [2, 5, 9, 7]. We noted the complications above related to μ CT imaging a whole vitrified liver, and further, there is not insignificant cost associated with preparing the quantity of nanoparticles required to perfuse a pig liver [Gao et al. *Advanced Science* 2020]. Therefore, for the proof of principle intended for this manuscript, we believe that it is adequate to independently demonstrated (i) we can vitrify and rewarm liter scale solutions with stock IONPs, and (ii) we can successfully vitrify whole livers. In addition, our current RF coil (~ 13 cm diameter) cannot be used to nanowarm these ~1L liver systems (~20 cm horizontal dimension). Thus, we elected to focus solely on whether or not these large liver systems can vitrify. Having successfully achieved this new milestone, we can move to designing larger RF coils to nanowarm these systems in the future.

Added material:

Fig. S15: Photographs and μ CT images of livers. Visual photos of 100-200mL segments of A. Frozen vs. B. vitrified liver. μ CT images showing Hounsfield units (HU) in the C. frozen vs. D. vitrified liver. A small section (\sim 200mL) of the \sim 1L vitrified liver (as from Figure 4 in main) was scanned to fit in the μ CT setup. Vitrified liver displays uniform HU intensity whereas frozen liver shows non-uniform and lower HU. Note that lower HU intensity indicates the presence of ice in a CPA loaded, but intentionally frozen liver (A) which we did not include in Fig. S14 inset Table. Both frozen and vitrified livers were fractured intentionally to create smaller segments which can fit in the μ CT setup. Two fractured segments of frozen liver are included to create similar volume for comparison to the vitrified liver segment. Note that some ice formed on the surface of the vitrified liver segment during the handling and imaging process (B). Furthermore, the edge of the vitrified liver (D) shows a smaller HU due to common edge artifacts, such as the beam hardening effect.

Revised Figure 4:

Fig. 4: Photos of a representative porcine liver a, d (left) before ($T = 4^{\circ}\text{C}$) and b, e (right) after vitrification ($T = -150^{\circ}\text{C}$). The pattern in the vitrified liver was due to the cryobag placement on a supporting mesh in the control rate freezer (CRF) (see Fig. S7B). The cryobag was removed for the vitrified liver photo to reduce glare and get a clear photo. Rightmost photos of a center cross-section of vitrified bisected pig liver. Blue (*) indicates perihilar tissue, fatty tissue, and the cryobag. Note also surface ice condensation that occurred between storage and photography in the bottom bisected liver (**).

8. SI, line 157: “perfused CPA long enough to achieve >96% equilibration in tissue to demonstrate physical vitrification success (verified using the effluent reflective index).” Please provide more evidence to support this statement.

We have added 2 plots namely RI vs. CPA concentration calibration curve and calculated CPA concentration from measured RI values vs. time during perfusion in Figure S5 along with additional explanation in supplementary section as reproduced below. Furthermore, we have clarified that effluent is equilibrated to >95% and not necessarily tissue.

Added Material:

“For this study, we perfused CPA long enough to achieve >95% equilibration for effluent to demonstrate physical vitrification success (verified using the effluent refractive index). Refractive index (RI) was first measured for a few diluted CPA concentrations (0%-1xEC, 90, 95%v/v of 40%v/vEG+0.6M Sucrose) since the full concentration CPA was outside the refractometer detection range to create. The resulting CPA concentration vs RI calibration curve as shown in Fig. S5B. Effluent was collected during the final step at interval of 15 mins and RI was measured until the refractometer readout became higher than detection limit which corresponds to >95% dilution of last step CPA concentration i.e. 40%EG+0.6M Sucrose (Fig. S5C).”

Revised Figure S5:

Fig. S5: Perfusion of a ~0.8L porcine liver with 40%EG+0.6MSucrose: **A** Photo of the liver after cannulation via the portal vein. **B** Calibration curve of measured refractive index (RI) vs. EG concentration. **C** Step loading of CPA (concentration vs. time). Black circles represent the perfused concentration in each time step whereas green circle represents the measured CPA conc. over time calculated via RI. Note that the first 15 minutes correspond to 0%EG (i.e., Euro Collins-carrier solution only). The effluent CPA concentration was calculated using the measured RI and calibration curve which is plotted as open circles in C.

Minor concerns:

1. Considering that figures serves as the primary illustration in the main text, the image quality of Fig.2 can be further improved.

We have increased the quality of Figure 2. Additional photos of vitrified CPAs are presented in supplementary Figures S10 and 11.

2. Fig.3: The picture is not clear, the shaded part of Fig.3 b,c affects the clarity.

We have removed the shaded box in Fig. 3b, c to increase the clarity.

References

[1] Gangwar, Lakshya, et al. "Perspective: A Guide to Successful ml to L Scale Vitrification and Rewarming." *CryoLetters* 43.6 (2022): 303-315.

[2] Sharma, Anirudh, et al. "Vitrification and nanowarming of kidneys." *Advanced Science* 8.19 (2021): 2101691.

[3] Fahy, Gregory M., et al. "Physical and biological aspects of renal vitrification." *Organogenesis* 5.3 (2009): 167-175.

[4] Wowk, Brian, et al. "27 MHz constant field dielectric warming of kidneys cryopreserved by vitrification." *Cryobiology* 115 (2024): 104893.

[5] Han, Zonghu, et al. "Vitrification and nanowarming enable long-term organ cryopreservation and life-sustaining kidney transplantation in a rat model." *Nature communications* 14.1 (2023): 3407.

[6] Kuleshova, L. L., et al. "Vitrification of encapsulated hepatocytes with reduced cooling/warming rates." *CryoLetters* 25.4 (2004): 241-254.

[7] Sharma, Anirudh, et al. "Cryopreservation of whole rat livers by vitrification and nanowarming." *Annals of biomedical engineering* 51.3 (2023): 566-577.

[8] Fahy, Gregory M., et al. "Improved vitrification solutions based on the predictability of vitrification solution toxicity." *Cryobiology* 48.1 (2004): 22-35.

[9] Gao, Zhe, et al. "Vitrification and rewarming of magnetic nanoparticle-loaded rat hearts." *Advanced materials technologies* 7.3 (2022): 2100873.

[10] Steif, Paul S., Daniel A. Noday, and Yoed Rabin. "Can thermal expansion differences between cryopreserved tissue and cryoprotective agents alone cause cracking." *CryoLetters* 30.6 (2009): 414-421.

[11] Rabin, Yoed, and Joseph Plitz. "Thermal expansion of blood vessels and muscle specimens permeated with DMSO, DP6, and VS55 at cryogenic temperatures." *Annals of Biomedical Engineering* 33 (2005): 1213-1228.

[12] Eisenberg, David P., Michael J. Taylor, and Yoed Rabin. "Thermal expansion of the cryoprotectant cocktail DP6 combined with synthetic ice modulators in presence and absence of biological tissues." *Cryobiology* 65.2 (2012): 117-125.

[13] Wowk, Brian, and Gregory M. Fahy. "Inhibition of bacterial ice nucleation by polyglycerol polymers." *Cryobiology* 44.1 (2002): 14-23.

Thank you for continuing to consider our manuscript. Accordingly, we have undertaken the following changes:

1. We have refined the wording throughout to emphasize that this is a physical not a biological study. We have highlighted the changes in the revised versions.
2. We have refined the wording to emphasize that this is a proof-of-concept study in the abstract and introduction of the manuscript.
3. We have removed the statement: 'Combined with our group's related demonstration of a robust protocol for long-term cryopreservation, rewarming, and successful transplantation of a functional preserved rat kidney, these findings lay the groundwork for achieving clinical organ banking'.
4. We have indicated the limitations of this work in the final paragraph of the discussion section as reproduced below.

“In the future, several limitations of this work will need to be addressed. No biological assessment of the liver was performed, and further optimization and development of the perfusion protocol will be required to minimize CPA toxicity and achieve consistent physical and biological success. For example, we assumed that CPA will fully equilibrate in tissue, but, as noted, tissue equilibration is often only in the range of ~92-94% after organ perfusion [15, 34]. Nevertheless, experimental work on such dilutions of VMP and VS55 in tissue show similar vitrifiability to 100% CPA [48]. Furthermore, due to the limitations of the 120 kW RF coil diameter (Fig.S24), none of the vitrified livers in this study were nanowarmed. As this work is focused solely on assessing feasibility for physical vitrification and nanowarming success (i.e. no biological and functional assessment), the toxicity of specific CPAs in specific organs will be addressed elsewhere. Both physical and biological studies will be needed to ultimately achieve successful human scale organ banking.”

5. We have indicated the experiments that exceed the scope of this physical proof-of-concept study in our responses to the reviewers.
 - a. Biological assessments such as cell viability assays, structural integrity analysis, and functional metrics.
 - b. Perfusion optimization (requires physical and biological optimization).
 - c. Rewarming of porcine livers.

Reviewer #2 (Remarks to the Author):

Thank the authors for carefully addressing my comments. I still feel the title of the manuscript is misleading and the last four words "to enable organ cryopreservation" in the title should be replaced with "of cryopreservation solutions", as the authors clarified that they did not warm the vitrified porcine liver in this work. Other than this, the work is acceptable for publication in this journal.

Based on both reviewer and editor comments we have modified the title. We believe the modified title addresses all concerns:

“Physical vitrification and nanowarming at liter-scale CPA volumes: Toward organ cryopreservation.”

Reviewer #3 (Remarks to the Author):

The revised manuscript demonstrates progress through the addition of thermal simulations and repeated liver vitrification experiments. However, several critical issues require clarification to strengthen the scientific rigor and resolve reader confusion.

Specific Comments:

1. Title-Abstract Discrepancy & Experimental Scope Clarification

The title claims "Physical vitrification and nanowarming at liter volume to enable organ cryopreservation", yet the abstract explicitly states "...we successfully vitrified although did not rewarm...". This inconsistency creates confusion regarding the study's achievements. While the authors successfully demonstrated vitrification and nanowarming in CPA solutions (1L to 3L), organ (226ml to 752ml) cryopreservation remains incomplete due to the absence of organ rewarming (a complete cryopreservation includes cooling process and warming process).

We agree with the reviewer and have refined the wording throughout to emphasize that this is a physical proof-of-concept study only and not a biological study. We have changed the title accordingly as you requested (in your comment below) to:

"Physical vitrification and nanowarming at liter-scale CPA volumes: Toward organ cryopreservation."

The justification for not rewarming organs - "the size of RF coil is smaller than organ dimensions" - conflicts with experimental evidence:

- The 3L CPA group (cryobag dimensions: 19×30×10.5 cm) underwent successful nanowarming, yet the 0.752L organ (max dimension ~20 cm) was deemed incompatible with the RF coil (13.3 cm diameter limit).
- Similarly, the 1L CPA cryobag (19.5×14.5×6.5 cm) accommodated nanowarming, while a 0.752L organ (smaller than 1L cryobag dimensions) could not.

This discrepancy requires clarification: Why can nanowarming be achieved in larger-volume CPA systems but not in smaller-volume organs? A quantitative comparison of spatial constraints (e.g., organ geometry vs. RF coil configuration) should be provided.

The 120 kW RF coil geometry is a cylinder with a diameter of 13.3 cm and a uniform rewarming volume of up to 2.5 L (Figs. 7, S21). The widths of the larger 2 L and 3 L bags used for the largest vitrification volumes shown in Fig. 2 exceed this diameter and will not fit inside the RF coil. For this reason, vitrification and nanowarming were not always performed with the same volumes, as detailed in Table S3. Furthermore, Fig. S21 presents the actual dimensions of the cryobags used for nanowarming volumes of 0.5 L, 1 L, and 2 L.

To address this, we have added clarifications in the Results section of the manuscript:

“Different cryobags than those used in the vitrification studies were used to fit in with the workable volume of our 120 kW RF coil...”

“Note that the vitrified porcine liver was not nanowarmed due to RF coil dimension restriction (diameter of coil being smaller than horizontal flat liver dimensions; also see Fig. S24).”

To further clarify, we have now also included Supplementary Fig. S24, which illustrates the geometric constraints of the RF coil in comparison with the sample geometries, including organs and CPA cryobags. Additionally, the physical dimensions of the RF coil and cryobags are also provided in Tables S3 and S5, respectively, to emphasize the practical limitations encountered in the study.

2. Inaccurate Organ Volume Description

The manuscript describes porcine livers as “~0.6-1L total volume”, yet the actual experimental volumes (Fig. S14) range from 0.226L to 0.752L (the actual volume of porcine livers used in the experiment is 750 ml, 650 ml, 752 ml, 226ml and 250 ml, respectively). The “~0.6-1L” claim is misleading and should be revised to reflect the true data range (0.226-0.752L). If smaller livers (<0.6L) were intentionally excluded from analysis, this must be explicitly stated with justification.

We regret this misunderstanding and have added the following statement in the abstract to better clarify the experimental details and results:

“~0.6-1L total volume; ~0.23-0.75L organ volume”

This emphasizes that the total volume includes both the liver and the surrounding cryoprotective agent (CPA) solution. This distinction is also explicitly stated in both the Methods and Results sections which refer to Fig. S14 – Table inset.

3. Figure 4 Caption & Annotation Issues

- Missing labels: Captions for Fig. 4c and 4f are absent.
- Ambiguous annotations:
 - The statement "Blue (*) indicates perihilar tissue, fatty tissue, and the cryobag" is confusing. The cryobag appears to be represented by bubbles between the organ and bag wall; a distinct symbol (e.g., †) should be used for clarity.
 - The description of "surface ice condensation (**)" lacks spatial context. The bisected liver region should be clearly marked in Figs. 4a,b,d,e. And the photography taking also need to maintain low-temperature condition to avoid extra ice formation.

As suggested, we have added captions for Figs. 4c and 4f and modified the figure to include added symbols for fatty tissue, cryobag, and surface ice condensation.

Fig. 4: Photos of a representative porcine liver a, d (left) before ($T = 4^{\circ}\text{C}$) and b, c, e, f (right) after vitrification ($T = -150^{\circ}\text{C}$). The pattern in the vitrified liver was due to the cryobag placement on a supporting mesh in the control rate freezer (CRF) (see Fig. S7B). The cryobag was removed for the vitrified liver photo to reduce glare and get a clear photo. Rightmost photos c, f of a center cross-section of vitrified bisected pig liver. Yellow dashed line in b, e represents the location of an orthogonal bisection in liver. Blue (*) indicates perihilar and fatty tissue whereas blue (‡) denotes the cryobag. Blue (†) denotes some surface ice condensation that occurred between storage and photography in the bottom bisected liver.

4. Optimization of the CPA perfusion loading protocol

Ice formation occurred during vitrification in organ sample 2 and 3 (Fig.S14), the author explained that “further optimization of the CPA perfusion loading protocol may be required which was not the focus of this manuscript”. Since CPA perfusion and distribution in organ can directly affect vitrification and rewarming outcomes, and it is also a physical problem. Optimization of the CPA

perfusion loading protocol is suggested, or the authors could prove that a trace amount of ice at the periphery will not affect the function of the organ by biological data.

The perfusion protocol applied to the liver in this study was intended as a proof-of-concept to demonstrate the feasibility of loading a large organ with sufficient CPA to physically vitrify. While we maintain that our data successfully demonstrates that porcine livers can vitrify, we fully acknowledge that further optimization and development are needed to achieve robust, repeatable outcomes. More specifically, while several livers were vitrified as hoped, some did form a small amount of ice as shown in Fig. S14. This may be attributed to biological variability among livers, potentially leading to inhomogeneous CPA perfusion, rather than being solely a consequence of suboptimal perfusion protocol parameters. This explanation has now been included in the revised supplementary material (caption of Fig. S14) as a possible contributing factor to the observed ice formation.

Furthermore, to address this, we have now explicitly stated in the main and supplementary text that this perfusion protocol represented a preliminary feasibility study, and that further refinement will be necessary to ensure consistent physical and biological success.

Main text (discussion section):

“In the future, several limitations of this work will need to be addressed. No biological assessment of the liver was performed, and further optimization and development of perfusion protocol will be required to minimize CPA toxicity and achieve consistent physical and biological success.”

Supplementary text (porcine liver perfusion section):

“Note that this perfusion protocol was adapted to provide a proof-of-concept demonstration of physical vitrification success in a large organ. Since biological assessment was outside the scope of this work, future work would need to further optimize this protocol to reduce CPA toxicity.”

In short, we see the perfusion protocol optimization as requiring both physical and biological assessments and therefore is outside the scope of this work.

5. Biological Validation Deficiency

While Reviewer #2 and #3 previously requested basic biological assessments (tissue viability, integrity), the authors only provide visual inspection of 100-200mL segments (Fig. S15). Given the RF coil limitation for whole-organ rewarming, minimum validation could include:

- Cell viability assays (e.g., live/dead staining) on rewarmed tissue segments
- Structural integrity analysis (H&E staining) to assess CPA toxicity and ice damage

- Functional metrics (e.g., ATP levels) for sub-samples

Even preliminary biological data would significantly strengthen the claim of "feasibility for organ cryopreservation", as physical vitrification alone cannot confirm biospecimen functionality.

We agree with the reviewer that we do not have biological validation data which are outside the scope of this physical study. We have refined the wording throughout to clarify this emphasis, including modifying the title as suggested by the reviewer to:

"Physical vitrification and nanowarming at liter-scale CPA volumes: Toward organ cryopreservation"

Additional Suggestions:

- Consider renaming the title to "Physical vitrification and nanowarming at liter-scale CPA volumes: Toward organ cryopreservation" to better align with demonstrated results.

Modified as suggested.

- Include a schematic of RF coil dimensions vs. organ/cryobag geometries in supplemental materials.

This revision can enhance clarity while preserving the original intent.

The dimensions of the RF coil are provided in Table S5 and Figure 5, while Table S3 lists the corresponding dimensions of the cryobags used for both CPA solutions and porcine liver samples. Additionally, the geometries of the vitrified CPA cryobags are depicted in Figures 2, S10, and S11, while those for nanowarming are shown in Figure S21.

To further enhance clarity, we have included Fig. S24 (as reproduced below), a schematic illustration designed to facilitate direct comparison between the RF coil geometry and the sample geometries, including both organs and CPA cryobags.

Figure S24: Schematic of cryobag dimensions constrained by CRF and 120 kW RF Coil. Three geometries (labeled as 1, 2, 3) were tested for liter scale vitrification. The cryobag with IONP geometry (2) was the only group which was nanowarmed after vitrification whereas other geometries (1) - CPA only (results in Fig. 2) and (2) - porcine liver results in Fig. 4 were used for vitrification only. The liver was vitrified while placed horizontally (see Fig. S7) whereas all the CPA only volumes were vitrified hung vertically inside the CRF. Note that the range of dimensions correspond to varying sample volume (0.5, 1, 3L for vitrification only, 0.5, 1, 2L for nanowarming, and ~1L for porcine liver case). These dimensions represent cryobag dimensions after vitrification and the actual cryobag dimensions without any sample are listed in Table S3.

Reviewer #3 (Remarks to the Author):

After thorough evaluation of the authors' responses and revised manuscript, I maintain significant concerns regarding the scientific validity and novelty of this work. My objections are outlined below:

1.The authors repeatedly emphasize this as a "physical proof-of-concept" study, asserting that biological validation "exceeds the scope." Cryopreservation research ultimately aims to preserve "biological function". Physical vitrification/rewarming without any viability assessment (e.g., cell survival, structural integrity, or functional recovery) fails to demonstrate feasibility for organ cryopreservation.

We respectfully disagree with the reviewer and maintain that our physical study has merit as submitted.

2.As to the physical approach, there is a lack of methodological and material innovation. The study offers no significant advances over existing literature, nanowarming via AMF is well-established (e.g., Manuchehrabadi et al., Sci Transl Med, 2017; Sharma et al., Adv Sci, 2021). M22 CPA is not novel (Fahy et al., Cryobiology, 2004), and the authors implemented it without modifications to enhance biocompatibility or efficacy. The core work-screening CPA compatibility for large volume constitutes incremental engineering optimization, which is insufficient. If the focus is purely physical, this work belongs in specialized journals.

We respectfully disagree with the reviewer that we lack innovation. This work greatly exceeds the scale of all previous work we are aware of for both vitrification and rewarming.

3.Only 1 of 3 porcine livers (752 mL) achieved full vitrification, others (650 mL, 720 mL) showed ice formation (Fig. S14). The authors attribute this to "biological variability," but no data supports this claim. This inconsistency undermines the physical protocol's robustness.

We took pains to add more examples of vitrified livers since our second revision. The fact that we have vitrified 3 (750, 226, 250 mL - Fig.S14) out of 5 (total livers) is a clear evidence that vitrification of such large-scale organs is feasible. We believe that our data is the first to demonstrate this large scale physical vitrification success that we are aware of.

4.The authors claim rewarming organs was impossible due to RF coil constraints (diameter: 13.3 cm). However, it seems that ~200 mL liver could fit Cryobag #2 (~11×13×30 cm; Fig. S24), which was used for rewarming experiment. Rewarming tissue segments (e.g., 100–200 mL) to assess viability was technically feasible yet omitted.

We agree that rewarming of iron oxide loaded tissue segments would in principle be possible. However, we elected not to do this as due to the technical difficulties of IONP loading of tissue segments, and the fact that we already achieved our goal of showing that we can physically vitrify and rewarm at the large (liter) scales shown.

5.The response letter and manuscript contain grammatical errors (e.g., "Different cryobags than those used..." → should be "...that..."), reflecting insufficient revision rigor.

We thank the reviewer for these valuable comments. We have made these corrections, and have also proofread, and spell checked the article again carefully.